# Using paired serology and surveillance data to quantify dengue transmission and control during a large outbreak in Fiji

Adam J Kucharski[1,2]*, Mike Kama[3,4], Conall H Watson[1,2], Maite Aubry[5], Sebastian Funk[1,2], Alasdair D Henderson[1,2], Oliver J Brady[1,2], Jessica Vanhomwegen[6], Jean-Claude Manuguerra[6], Colleen L Lau[7], W John Edmunds[1,2], John Aaskov[8], Eric James Nilles[9], Van-Mai Cao-Lormeau[5], Stéphane Hué[1,2], Martin L Hibberd[10]

[1]Centre for the Mathematical Modelling of Infectious Diseases, London School of Hygiene and Tropical Medicine, London, United Kingdom; [2]Department of Infectious Disease Epidemiology, London School of Hygiene and Tropical Medicine, London, United Kingdom; [3]National Centre for Communicable Disease Control, Suva, Fiji; [4]University of the South Pacific, Suva, Fiji; [5]Unit of Emerging Infectious Diseases, Institut Louis Malardé, Tahiti, French Polynesia; [6]Institut Pasteur, Paris, France; [7]Research School of Population Health, Australian National University, Canberra, Australia; [8]Queensland University of Technology, Brisbane, Australia; [9]World Health Organization Division of Pacific Technical Support, Suva, Fiji; [10]Department of Pathogen Molecular Biology, London School of Hygiene and Tropical Medicine, London, United Kingdom

*For correspondence:
adam.kucharski@lshtm.ac.uk

**Abstract** Dengue is a major health burden, but it can be challenging to examine transmission and evaluate control measures because outbreaks depend on multiple factors, including human population structure, prior immunity and climate. We combined population-representative paired sera collected before and after the 2013/14 dengue-3 outbreak in Fiji with surveillance data to determine how such factors influence transmission and control in island settings. Our results suggested the 10–19 year-old age group had the highest risk of infection, but we did not find strong evidence that other demographic or environmental risk factors were linked to seroconversion. A mathematical model jointly fitted to surveillance and serological data suggested that herd immunity and seasonally varying transmission could not explain observed dynamics. However, the model showed evidence of an additional reduction in transmission coinciding with a vector clean-up campaign, which may have contributed to the decline in cases in the later stages of the outbreak.
DOI: https://doi.org/10.7554/eLife.34848.001

## Introduction

In recent years, the reported incidence of dengue has risen rapidly. In the Asia-Pacific region, which bears 75% of the global dengue disease burden, there are more than 1.8 billion people at risk of infection with dengue viruses (DENV) (*World Health Organization, 2009*). Increased air travel and urbanisation could have contributed to the geographic spread of infection (*Gubler, 1998*; *Simmons et al., 2012*), with transmission by mosquitoes of the *Aedes* genus, including *Aedes aegypti* and *Aedes albopictus* (*Halstead, 2007*). DENV has four serotypes circulating, with infection conferring lifelong protection against the infecting serotype and short-lived protection against the

**eLife digest** Dengue fever – a disease spread by mosquitos – causes large outbreaks in Asia and the South Pacific islands. Health agencies often try to reduce the spread of the disease by removing mosquito breeding grounds, like old tires and containers that may hold standing water. But it can be difficult to tell whether these preventive measures work because dengue transmission depends on many factors, including the weather and how many people had developed immunity because of previous infections.

A common way to study patterns of infection and immunity is to collect blood samples from a subset of the population before and after an outbreak. Unfortunately, large dengue outbreaks occur sporadically on islands, making it hard to set up a study like this ahead of an outbreak. During 2013 and 2014, there was a major dengue outbreak in Fiji, with over 25,000 suspected cases reported. In response, the government introduced a nationwide mosquito clean-up campaign. As luck would have it, a group of researchers had collected blood samples immediately before the outbreak for an unrelated study of typhoid fever and leptospirosis.

Now, Kucharski et al. – who include the researchers who collected those pre-outbreak blood samples – show that the clean-up campaign coincided with a reduction in transmission of the disease. Participants whose blood was collected before the dengue outbreak were invited to provide another blood sample after the dengue outbreak. This allowed Kucharski et al. to identify individuals who had already developed immunity to dengue before the outbreak and those who were likely infected during the outbreak.

Comparing blood samples taken before and after the outbreak revealed that children and teenagers between the ages of 10 and 19 had the greatest risk of infection during the outbreak. No other demographic or environmental factors were strongly linked to the likelihood of infection. Computer models using the data also showed that the clean-up efforts could explain the reduced dengue transmission during the outbreak. These findings suggest that studying immunity against dengue can lead to a better understanding of disease transmission. This may help health agencies to gauge the effects of efforts to control this disease, and possibly forecast future outbreaks.
DOI: https://doi.org/10.7554/eLife.34848.002

others (*Sabin, 1952*; *Guzmán and Kourí, 2002*). Although four serotypes of DENV may co-circulate in South East Asia, only one serotype circulates in most of the South Pacific islands at any point in time (*Cao-Lormeau et al., 2014*; *Li et al., 2010*).

Between November 2013 and July 2014, a major outbreak caused by DENV-3 occurred in Fiji, with more than 25,000 suspected cases reported (*Figure 1A*). Prior to the 2013/14 outbreak, there were eleven outbreaks of dengue recorded in Fiji, involving serotypes 1, 2 and 4 (*Table 1*). Most cases in 2013/14 occurred on Viti Levu, the largest and most populous island. This is administratively divided into the Central Division, which includes the port-capital Suva, and Western Division, which contains the urban centres of Lautoka and Nadi, where Fiji's major international airport is located. Dengue transmission in Central and Western Divisions is likely to be driven mostly by the *Aedes aegypti* vector, with *Aedes albopictus* most abundant in the Northern Division. *Aedes polynesiensis* and *Aedes pseudoscutellaris* are also present in all divisions (*Maguire et al., 1971*; *Prakash et al., 2001*). In response to the 2013/14 outbreak, considerable resources were dedicated to implementing control measures, including a nationwide vector clean-up campaign between 8th and 22nd March 2014 (*Break Dengue, 2014*). As well as media coverage and distribution of flyers to raise awareness about dengue prevention and protection, a major operation was put in place to remove rubbish that could act as egg laying habitats for mosquitoes. In total, forty-five tonnes of tyres and twenty-five tonnes of other containers were removed during this period.

Large dengue outbreaks can place a substantial public health burden on island populations (*Fagbami et al., 1995*; *Sharp et al., 2014*). However, understanding the dynamics of infection and evaluating the impact of vector control measures remains challenging. There is a limited evidence base for control measures even in controlled trials (*Bowman et al., 2016*; *Heintze et al., 2007*), and post-outbreak evaluation is hindered by the fact that the size and duration of major outbreaks can be influenced by several factors, including population immunity, human movement, seasonal

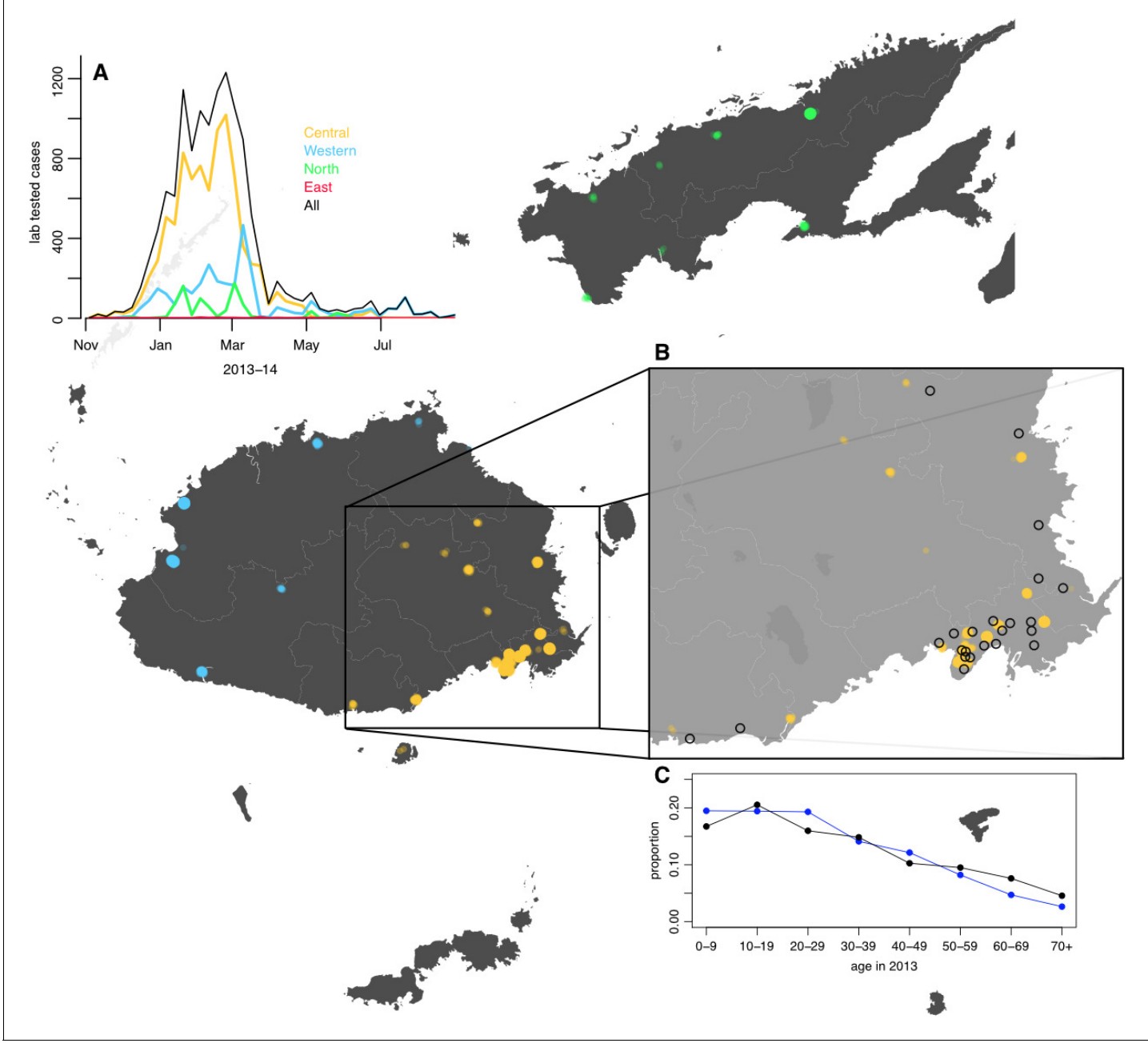

**Figure 1.** Geographical distribution of weekly lab tested suspected dengue cases in Northern (green), Western (blue) and Central (yellow) divisions between 27th October 2013 and 1 st July 2014. Points on the maps show locations of cases arranged by health centre they reported to; these are plotted with jitter and transparency to show concentrations of cases. (**A**) Weekly reported case totals for Northern, Western, Central and Eastern divisions. (**B**) Serosurvey study locations. Black circles show the 23 study clusters included in the analysis. (**C**) Age distribution of Central Division in the 2007 census (blue line) and ages of serosurvey participants in 2013 (black line).

DOI: https://doi.org/10.7554/eLife.34848.003

variation in transmission, and proportion of people living in urban, peri-urban and rural communities. In Fiji, dengue outbreaks typically occur during the wetter, warmer season between December and July, when vectors are most abundant (*Goettel et al., 1980*). Although surveillance data can provide broad insights into arbovirus transmission patterns (*Cuong et al., 2011*; *Funk et al., 2016*; *van Panhuis et al., 2015*), and cross-sectional serosurveys can be used to measure contemporary levels of immunity (*Aubry et al., 2015*; *Ferguson et al., 1999*; *Maguire et al., 1974*;

**Table 1.** Reported dengue outbreaks in Fiji between 1930–2014.

Two studies (*Fagbami et al., 1995*; *Maguire et al., 1974*) also included a post-outbreak serosurvey in Central Division. *There is also evidence of DENV-3 circulation during this period (*Singh et al., 2005*).

| Year | Main serotype | Reported cases | Seroprevalence | Source |
|------|---------------|----------------|----------------|--------|
| 1930 | ? | Thousands | | (*Maguire et al., 1971*) |
| 1944-5 | 1 | Thousands | | (*Reed et al., 1977*) |
| 1971-3 | 2 | 3413 | 26% (Suva) | (*Maguire et al., 1974*) |
| 1974-5 | 1 | 16,203 | | (*Reed et al., 1977*) |
| 1980 | 4 | 127 | | (*Fagbami et al., 1995*) |
| 1981 | 1 | 18 | | (*Kiedrzynski et al., 1998*) |
| 1982 | 2 | 676 | | (*Kiedrzynski et al., 1998*) |
| 1984-6 | ? | 490 | | (*Fagbami et al., 1995*) |
| 1988 | ? | 22 | | (*Fagbami et al., 1995*) |
| 1989-90 | 1* | 3686 | 54% (Suva) | (*Fagbami et al., 1995*; *Waterman et al., 1993*) |
| 1997-8 | 2 | 24,780 | | (*World Health Organization, 2000*) |
| 2001-3 | 1 | ? | | (*Halstead, 2008*) |
| 2008 | 4 | 1306 | | (*PacNet Report, 2008*; *ProMED-mail, 2008*) |
| 2013-14 | 3 | 25,496 | | Fiji MOH |

DOI: https://doi.org/10.7554/eLife.34848.004

*Waterman et al., 1993*), characterising infection dynamics in detail requires cohort-based seroepidemiological studies (*Cuong et al., 2011*; *Reiner et al., 2014*), which can be difficult to implement in island settings where outbreaks are infrequent and difficult to predict.

Immediately before the 2013/14 dengue outbreak in Fiji, a population-representative serological survey had been conducted to study leptospirosis and typhoid (*Lau et al., 2016*). To investigate patterns of dengue infection in 2013/14, we followed up participants from this survey in Central Division, to obtain a set of paired pre- and post-outbreak serological samples (see Materials and methods). We tested the paired samples for anti-DENV IgG antibodies using ELISA and a recombinant antigen-based microsphere immunoassay (MIA), and combined these data with dengue surveillance data to compare possible explanations for the outbreak dynamics. We measured age-specific and spatial patterns of infection and reported disease, and tested whether there were demographic and environmental risk factors associated with infection. Having characterised factors shaping individual-level infection risk, we used a Bayesian approach to fit a transmission dynamic model to both the serological survey and surveillance data in order to estimate the contribution of climate and control measures to the decline in transmission observed in 2014.

## Results

The pre- and post-outbreak serological survey included 263 participants from the Central Division, with age distribution of these participants consistent with the population distribution (*Figure 1B–C*). We found that 58.6% of participants (154/263) were ELISA seropositive to at least one DENV serotype in late 2013. Two years later, in October/November 2015, this had risen to 74.5% (196/263). Additional serotype-specific MIA tests confirmed that the largest rise in seroprevalence in Central Division was against DENV-3, from 33.1% to 53.2% (*Table 2*), consistent with the majority of RT-PCR-confirmed samples during the outbreak being of this serotype.

To characterise patterns of infection between 2013 and 2015, we first considered individual-level demographic, behavioural and environmental factors. Using a univariable logistic regression model, we compared seroconversion determined by ELISA with questionnaire responses about household environment and health-seeking behaviour (*Table 3*). The factors most strongly associated with seroconversion between 2013–15 among initially seronegative participants were: living in an urban or peri-urban environment (odds ratio 2.18 [95% CI: 0.953–5.11], p=0.068); reporting fever in preceding

**Table 2.** Number of participants who were seropositive to DENV in 2013 and 2015 as measured by ELISA and MIA. MIA any DENV denotes participants who were MIA seropositive to at least one DENV serotype. 95% CI shown in parentheses.

| Test | N | 2013 | 2013 (%) | 2015 | 2015 (%) | Difference |
|------|---|------|----------|------|----------|------------|
| ELISA | 263 | 154 | 58.6% (52.3–64.6%) | 196 | 74.5% (68.8–79.7%) | 16% (11.8–21%) |
| MIA any DENV | 263 | 193 | 73.4% (67.6–78.6%) | 216 | 82.1% (77–86.6%) | 8.75% (5.62–12.8%) |
| MIA DENV-1 | 263 | 177 | 67.3% (61.3–72.9%) | 198 | 75.3% (69.6–80.4%) | 7.98% (5.01–11.9%) |
| MIA DENV-2 | 263 | 33 | 12.5% (8.8–17.2%) | 41 | 15.6% (11.4–20.5%) | 3.04% (1.32–5.91%) |
| MIA DENV-3 | 263 | 87 | 33.1% (27.4–39.1%) | 140 | 53.2% (47–59.4%) | 20.2% (15.5–25.5%) |
| MIA DENV-4 | 263 | 79 | 30.0% (24.6–36%) | 99 | 37.6% (31.8–43.8%) | 7.6% (4.71–11.5%) |

DOI: https://doi.org/10.7554/eLife.34848.005

two years (odds 2.94 [1.08–8.38], p=0.037); and visiting a doctor with fever in the preceding two years (odds 3.15 [1.06–10.10], p=0.043). Of the participants who seroconverted, 10/38 (26.3% [13.4–43.1%]) reported visiting a doctor with fever in the preceding two years, 2/38 (5.26% [0.644–17.7%]) reported fever but did not visit a doctor, and 26/38 (68.4% [51.3–82.5%]) did not report fever (*Supplementary file 1A*).

As well as estimating infection by measuring seroconversion based on threshold values, we also considered the distribution of ELISA values. There was a noticeable right shift in this distribution between 2013 and 2015, with ELISA values increasing across a range of values (*Figure 2A*). As some of the individual-level changes in value between the two tests were likely to be due to measurement error (*Salje et al., 2014*), we fitted a mixture model to the distribution of changes in ELISA value (*Figure 2B*). We used a normal distribution with mean zero to capture measurement error, and a gamma distribution to fit rise that could not be explained by this error function. The fitted model suggested that a rise in value of at least three was more likely to be a genuine increase rather than measurement error, as shown by the dashed line in *Figure 2B*.

**Table 3.** Risk factors from a univariable logistic regression model. Sample population was all individuals who were seronegative in 2013 (n = 97), and outcome was defined as seroconversion as measured by ELISA. Number indicates total individuals with a given characteristic.

| Variable | Number | Odds ratio | p value |
|----------|--------|------------|---------|
| Demographic characteristics | | | |
| Age under 20 | 61 | 0.49 (0.21–1.13) | 0.10 |
| Male | 49 | 0.81 (0.36–1.84) | 0.62 |
| iTaukei ethnicity | 85 | 1.33 (0.39–5.32) | 0.66 |
| Environmental factors present | | | |
| Mosquitoes | 90 | 4.19 (0.68–80.85) | 0.19 |
| Used car tires | 61 | 1.80 (0.77–4.42) | 0.18 |
| Open water container(s) | 61 | 1.49 (0.64–3.58) | 0.37 |
| Air conditioning | 23 | 0.46 (0.15–1.26) | 0.15 |
| Blocked drains | 53 | 1.04 (0.46–2.38) | 0.92 |
| Location | | | |
| Urban or peri-urban | 50 | 2.18 (0.95–5.11) | 0.07 |
| Health seeking behaviour | | | |
| Fever in preceding 2 years | 20 | 2.94 (1.08–8.38) | 0.04 |
| Visited doctor with fever in preceding 2 years | 16 | 3.15 (1.06–10.13) | 0.04 |
| Household member visited doctor with fever in preceding 2 years | 9 | 2.08 (0.52–8.94) | 0.30 |

DOI: https://doi.org/10.7554/eLife.34848.006

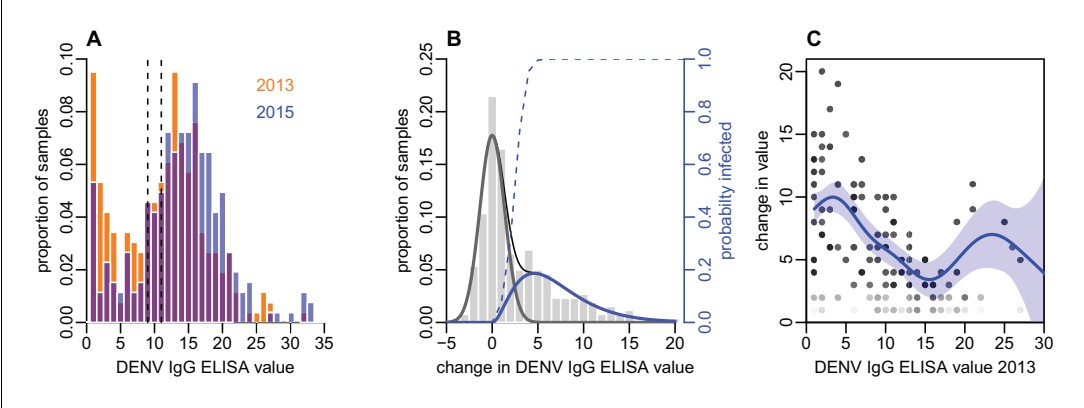

**Figure 2.** Distribution of ELISA values for anti-DENV IgG over time. (A) Distribution of values in 2013 and 2015. Orange bars show observed proportion of samples with each value in 2013; blue bars show proportions in 2015. Dashed lines show threshold for seronegativity and seropositivity. (B) Change in ELISA values between 2013 and 2015. Bars show distribution of values. Grey line shows estimated uncertainty in assay measurements; blue line shows estimated increase in value following the 2013–14 epidemic; thin black line shows overall fitted distribution (model $R^2$=0.93). Dashed line shows probability of infection for a given rise in value. (C) Relationship between value in 2013 and rise between 2013 and 2015, adjusting for probability of infection as shown in *Figure 2B*. Points show 1000 bootstrap samples of the data with replacement, with opacity of each point proportional to probability of infection. Blue line shows prediction from generalized additive model, with data points weighted by probability of infection; shaded region shows 95% CI (model $R^2$=0.31).

DOI: https://doi.org/10.7554/eLife.34848.007

To explore the relationship between the initial ELISA value and rise post-outbreak, given that an individual had been infected, we fitted a generalized additive model to the data and weighted each observation by the probability that a specific participant had been infected based on the dashed line in *Figure 2B*. By adjusting to focus on likely infections, we found a negative relationship between initial value and subsequent rise, with ELISA values near zero rising by around 10 units, but higher values exhibiting a smaller rise (*Figure 2C*). Using this approach, we also found strong evidence that self-reported symptoms were associated with larger rise in ELISA value, given likely infection. Using a logistic model with self-reported symptoms as outcome and change in value as dependent variable, adjusting for initial value and again weighting by probability of infection, we found that individuals who reported a fever in the preceding two years had a predicted rise in ELISA value that was 2.2 (95% 0.77–3.6) units higher than those who did not (p=0.003). Further, individuals who reported visiting a doctor with fever had a predicted value 3.3 (1.8–4.9) higher than others (p=0.0005).

Examining age patterns of seroprevalence, we found an increase in the proportion seropositive against DENV with age in both 2013 and 2015, and a rise in seroprevalence was observed in almost all age groups after the 2013/14 outbreak (*Figure 3A*). However, the high levels of seroprevalence in older age groups made it challenging to estimate age-specific probability of infection, because there was a relative lack of serologically naive individuals in these groups to act as a denominator (*Table 4*). We therefore again used rise in ELISA value as a correlate of infection, based on *Figure 2B*. As well as producing more precise estimates of infection risk in older groups (*Table 4*), this approach also suggested that individuals aged 10–19 years were most likely to be infected. This is in contrast to the surveillance data, which indicated the highest per capita level of reported disease was in the 20–29 age group (*Figure 3B*).

Next, we explored spatial patterns of infection in different communities. Previous studies have suggested that dengue outbreaks can spread outwards from urban hubs to more rural areas (*Cummings et al., 2004*; *Salje et al., 2017*). A similar spatial pattern was observed from the surveillance data during the early stages of the 2013/14 Fiji outbreak (*Figure 4A*). The first case was reported at Colonial War Memorial Hospital (CWM), Fiji's largest hospital located in central urban Suva, in the week ending 4th November 2013. The outbreak took 9 weeks to reach the furthest reporting point from CWM in Central Division, a health centre 51 km away by Euclidean distance (i.e. as the crow flies). We found limited association between Euclidean distance from CWM and

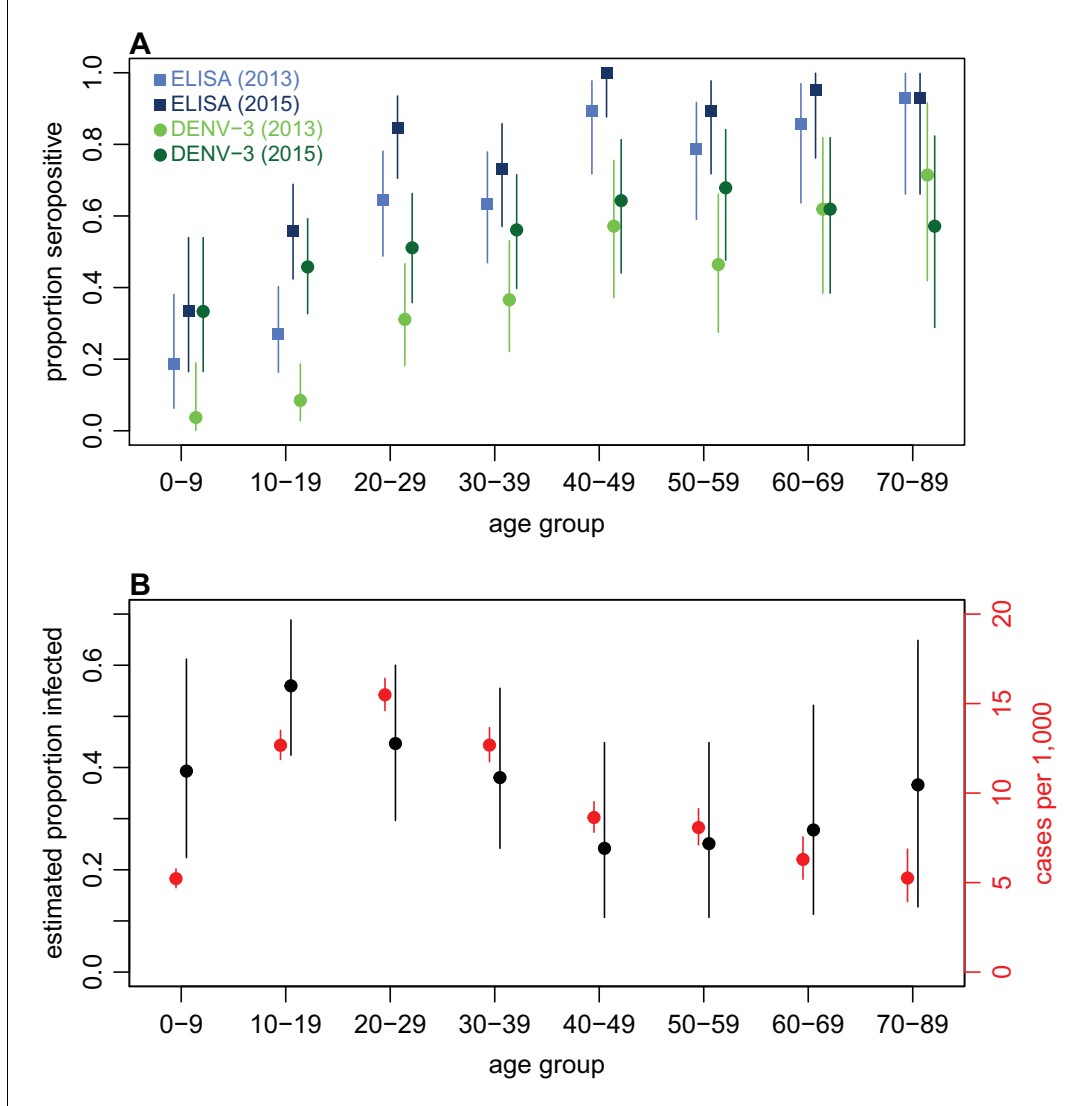

**Figure 3.** Age patterns of immunity and infection during 2013–15. (**A**) Proportion of each age group seropositive against DENV as measured by ELISA (blue squares) and DENV-3 by MIA (green circles). Lighter points show 2013 results, darker points show 2015; lines show 95% binomial confidence intervals. (**B**) Comparison of estimated age-specific infection and reported cases. Black points, estimated proportion infected based on ELISA rise indicated in *Figure 2B*; red points, cases reported per 1000 people in each age group; lines show 95% binomial confidence intervals.
DOI: https://doi.org/10.7554/eLife.34848.008

proportion of study cluster seropositive to DENV-3 in 2015 (*Figure 4B*): the Pearson correlation between ELISA seropositivity in each cluster and distance from CWM was $\rho= -0.12$ (p=0.59); for DENV-3 the correlation coefficient was $\rho= -0.46$ (p=0.03). However, we found no significant association between the Euclidean distance from CWM and proportion of cluster infected (*Figure 4C*). Pearson correlation between estimated proportion infected based on change ELISA value in each cluster and distance from CWM was $\rho= 0.22$ (p=0.30); for DENV-3 the correlation was $\rho= -0.36$ (p=0.09). We did find evidence of dengue seroconversion in every cluster, however, suggesting that the outbreak eventually spread throughout Central Division.

As we did not find strong evidence of individual or community-level heterogeneity in infection, we incorporated the surveillance data and paired serological survey data into mathematical models to test explanations for the observed outbreak dynamics at the division level. We considered three model variants: a simple age-structured model of vector-borne transmission dynamics; the same model structure, but with climate-driven variation in transmission; and a model with both climate-

**Table 4.** Estimated age-specific attack rates based on raw ELISA values, and seroconversion using ELISA cutoff.

Estimated proportions of infections were calculated from the total of the probabilities that each individual in that age group had been infected, based on change in ELISA values between 2013 and 2015 (*Figure 2B*). Binomial 95% confidence intervals are shown in parentheses.

| Age | N | Propn infected based on ELISA values | Seronegative | Seroconverted | Seroconverted (%) |
|---|---|---|---|---|---|
| 0–9 | 27 | 39.3% (22.2–59.3%) | 21 | 6 | 28.6% (11.3–52.2%) |
| 10–19 | 59 | 56% (44.1–67.8%) | 40 | 14 | 35% (20.6–51.7%) |
| 20–29 | 45 | 44.7% (31.1–60%) | 14 | 8 | 57.1% (28.9–82.3%) |
| 30–39 | 41 | 38% (24.4–53.7%) | 12 | 4 | 33.3% (9.92–65.1%) |
| 40–49 | 28 | 24.2% (10.7–39.3%) | 3 | 3 | 100% (29.2–100%) |
| 50–59 | 28 | 25.1% (10.7–42.9%) | 5 | 2 | 40% (5.27–85.3%) |
| 60–69 | 21 | 27.8% (9.52–47.6%) | 1 | 1 | 100% (2.5–100%) |
| 70+ | 14 | 36.6% (14.3–64.3%) | 1 | 0 | 0% (0–97.5%) |
| Total | 263 | 39.6% (33.8–45.6%) | 97 | 38 | 39.2% (29.4–49.6%) |

DOI: https://doi.org/10.7554/eLife.34848.009

driven variation in transmission and a potential additional reduction in transmission coinciding with the clean-up campaign in March 2014. When we jointly fitted the models to surveillance data and age-specific immunity, as measured by seropositivity to DENV-3 by MIA in 2013 and 2015, the model with both climate-driven variation in transmission and an additional transmission reduction performed best as measured by AIC and DIC (*Figure 5A–B* and *Table 5*). This additional reduction in transmission was modelled using a flexible additional sigmoidal transmission rate, and was constrained so that the midpoint of the decline occurred after the start of the campaign on 8th March 2014 (*Figure 5—figure supplement 1*); we estimated a reduction of 57% (95% CrI: 42–82%) in transmission that coincided with the clean-up campaign (*Figure 5C*). As the effective reproduction number was near the critical value of one when the clean-up campaign was introduced (*Figure 5D–E*), it suggests that the main contribution of control measures may have been to bring DENV-3 infections to sufficiently low levels for transmission to cease earlier. We obtained the same conclusions when ELISA rather than MIA seroprevalence was used to quantify immunity during model fitting (*Table 5* and *Figure 5—figure supplement 2*). It was noticeable that the model fitted to the ELISA data produced a qualitatively better fit to the surveillance data than the model fitted to MIA data. This was because the observed MIA values imposed a stronger constraint on the plausible range of model estimated seroprevalence (*Figure 5B* and *Figure 5—figure supplement 2B*), so in comparison the model fitted to ELISA data was able to attribute more of the slowdown in growth in the surveillance data during January/February to the accumulation of herd immunity.

Fitting to the DENV-3 MIA seroprevalence data, we estimated that the mean basic reproduction number, $R_0$, over the course of the year was 1.12 (95% CrI: 1.02–1.25), with a peak value of 1.87 (1.70–2.07) in January 2014 (*Table 6*). Posterior estimates are shown in *Figure 5—figure supplement 3* and correlation plots for the transmission rate parameters are shown in *Figure 5—figure supplement 4*. Accounting for stochastic variability in weekly case reporting, we estimated that 11% (1.1–39%) of infections were reported as laboratory-tested cases and 9.3% (1.1–39%) were reported as DLI cases. The estimated value of $R_0$ was larger for the model fitted to ELISA data, with a mean of 1.49 (1.35–1.69); this was the result of a larger proportion of the population assumed to be initially immune to infection.

As well as performing worse under AIC and DIC, the model with only climate-driven variation in transmission could not capture the overall shape of the surveillance data (*Figure 5—figure supplement 5*). The basic model, which had neither climate-driven variation in transmission nor an additional reduction in transmission, could not jointly reproduce both sets of data either (*Figure 5—figure supplement 6*). Fitting the basic model to the surveillance data alone, we could reproduce the observed incidence pattern under the assumption of a simple immunising epidemic. Specifically, the reported cases were consistent with an epidemic that declined as a result of depletion of the susceptible population (*Figure 5—figure supplement 7*). However, this basic epidemic model

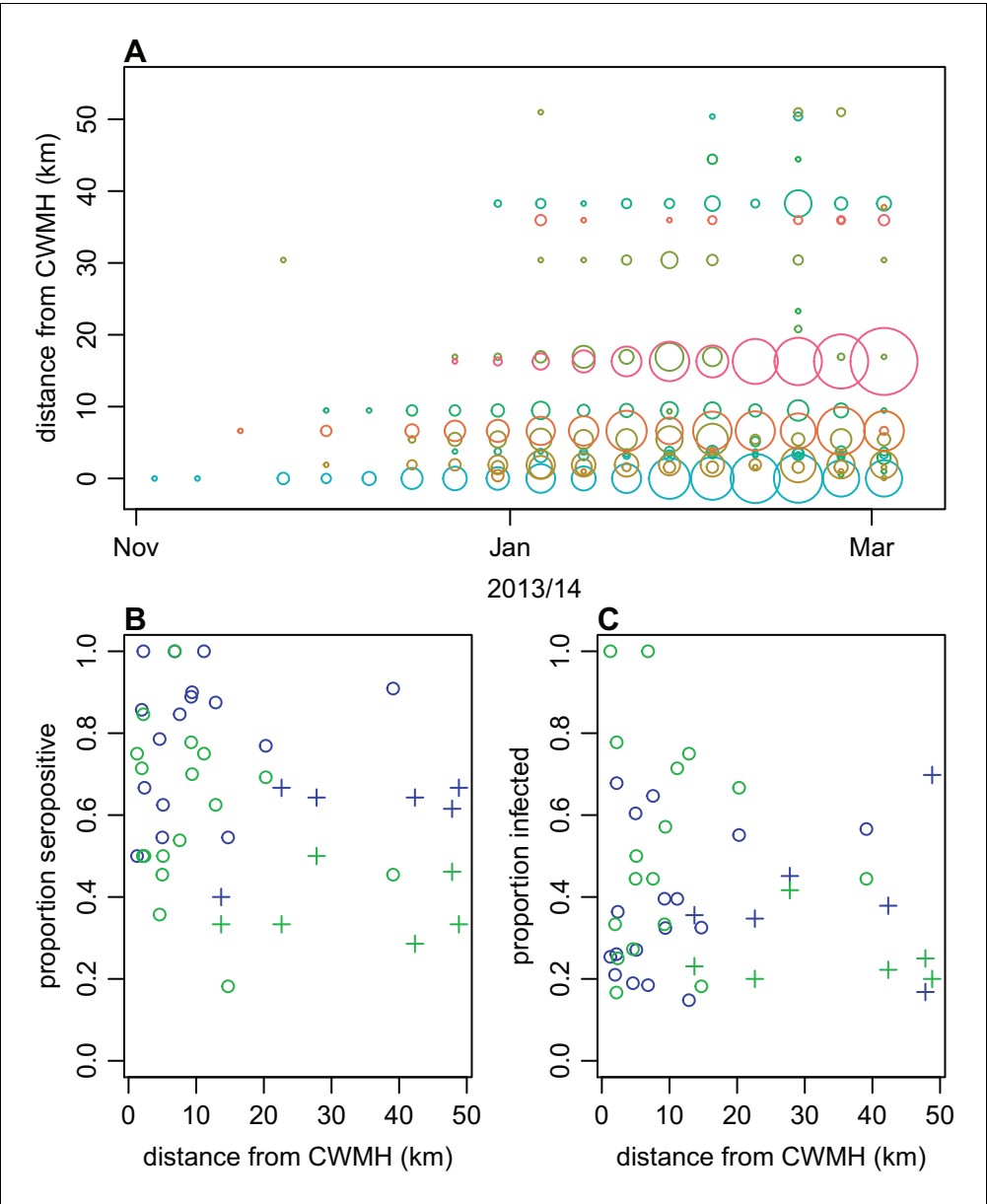

**Figure 4.** Spatial pattern of infection and immunity in Central Division. (**A**) Relationship between dengue cases reported by each health centre at the start of the outbreak and Euclidean distance from Colonial War Memorial Hospital (CWM) in Suva. Area of circle is proportional to number of cases reported in that week; each health centre is represented by a different colour. (**B**) Proportion seropositive in each serosurvey study cluster in 2015 vs Euclidean distance from CWM. Blue, ELISA data; green, MIA data; circles, urban or peri-urban clusters; crosses, rural clusters. (**C**) Proportion infected in each serosurvey study cluster vs Euclidean distance from CWM. Blue, estimate based on ELISA data, using adjustment in *Figure 2B*; green, seroconversion based on MIA for individuals who were initially seronegative; circles, urban or peri-urban clusters; crosses, rural clusters.

DOI: https://doi.org/10.7554/eLife.34848.010

underestimated the initial level of immunity and overestimated final immunity. A similar discrepancy between serological surveys and surveillance data has been noted in previous arbovirus modelling studies, albeit for ZIKV rather than DENV (*Funk et al., 2016*; *Kucharski et al., 2016*; *Champagne et al., 2016*).

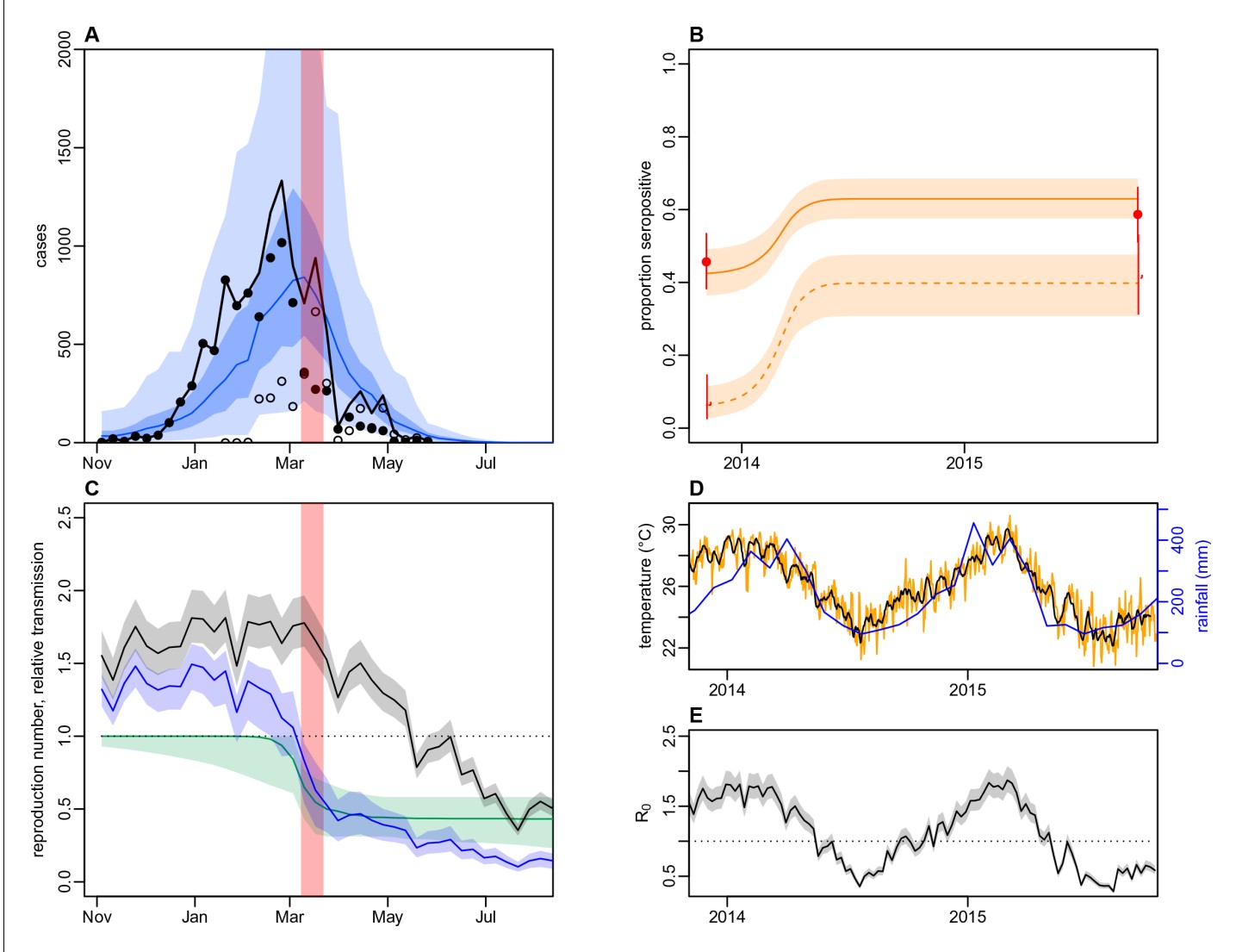

**Figure 5.** Impact of climate and control measures on DENV transmission during 2013/14, using a model jointly fitted to surveillance and serological data from Central Division. (A) Model fit to surveillance data. Solid black dots, lab tested dengue cases; black circles, DLI cases; black line, total cases. Blue line shows median estimate from fitted model; dark blue region, 50% credible interval; light blue region, 95% CrI; red region shows timing of clean-up campaign. (B) Pre- and post-outbreak DENV immunity. Red dots show observed MIA seroprevalence against DENV-3 in autumn 2013 and autumn 2015; hollow dots, under 20 age group; solid dots, 20+ age group; lines show 95% binomial confidence interval. Dashed orange line shows model estimated rise in immunity during 2013/14 in under 20 group; solid line shows rise in 20+ group; shaded region shows 95% CrI. (C) Estimated variation in transmission over time. Red region, timing of clean-up campaign; green line, relative transmission as a result of control measures. Black line, basic reproduction number, $R_0$; blue line, effective reproduction number, $R$, accounting for herd immunity and control measures. Shaded regions show 95% CrIs. Dashed line shows the $R = 1$ herd immunity threshold. (D) Average monthly rainfall (blue lines) and daily temperature (orange line, with black line showing weekly moving average) in Fiji during 2013–15. (E) Change in $R_0$ over time. Shaded regions show 95% CrIs.

DOI: https://doi.org/10.7554/eLife.34848.011

The following figure supplements are available for figure 5:

**Figure supplement 1.** Illustration of model variation in transmission as a result of climate and control.

DOI: https://doi.org/10.7554/eLife.34848.012

**Figure supplement 2.** Dynamics of DENV transmission during 2013/14, using a model jointly fitted to surveillance and ELISA serological data from Central Division.

DOI: https://doi.org/10.7554/eLife.34848.013

**Figure supplement 3.** Posterior parameter estimates for the model with climate and control measures.

DOI: https://doi.org/10.7554/eLife.34848.014

**Figure supplement 4.** Correlation between posterior distributions of transmission rate parameters.

*Figure 5 continued on next page*

*Figure 5 continued*

DOI: https://doi.org/10.7554/eLife.34848.015

**Figure supplement 5.** Dynamics of DENV transmission during 2013/14, using a model jointly fitted to surveillance and MIA serological data from Central Division, with only climate-based variation in transmission.

DOI: https://doi.org/10.7554/eLife.34848.016

**Figure supplement 6.** Dynamics of DENV transmission during 2013/14, using a model jointly fitted to surveillance and MIA serological data from Central Division, without time-varying transmission.

DOI: https://doi.org/10.7554/eLife.34848.017

**Figure supplement 7.** Dynamics of DENV transmission during 2013/14, using a model fitted only to surveillance data from Central Division, without time-varying transmission.

DOI: https://doi.org/10.7554/eLife.34848.018

## Discussion

We analysed surveillance reports and serological survey data to examine the dynamics of a major 2013/14 dengue outbreak in Fiji. Owing to the sporadic and unpredictable nature of dengue outbreaks in the Pacific (*Cao-Lormeau et al., 2014*), it is rare to have access to paired population-representative sera collected before and after such an epidemic. Comparing surveillance and serological survey data made it possible to investigate the relationship between observed reported cases and the true attack rate and quantify the relative role of climate, herd immunity and control measures in shaping transmission.

Analysis of detailed serological data provided insights into age-specific patterns of infection that would not be identified from seropositivity thresholds alone. We estimated the highest infection rate was in the 10–19-year-old age group, whereas proportionally the most reported cases were in the 20–29-year-old group. The apparent disparity between reported cases and infections estimated from the serological survey may be the result of secondary DENV infections causing more severe clinical disease and therefore increasing the likelihood of seeking medical care (*OhAinle et al., 2011*). The ELISA results suggested that fewer than 50% of individuals under age 20 had experienced DENV infection in 2013 (*Figure 3A*), which means an infection during the 2013/14 outbreak in this group was more likely to be primary than secondary. In contrast, the majority of 20–29 year olds already had evidence of infection in 2013, and hence 2013/14 outbreak would have generated relatively more secondary or tertiary infections in this group. In addition, if age-specific infection rates are indeed higher in younger groups, it means that estimating population attack rates based on the proportion of seronegative individuals infected may over-estimate the true extent of infection. Focusing on the seronegative subset of the population leads to children being over-sampled, which in our data inflates attack rate estimates by around 10% compared to estimates based on change in ELISA value (*Table 4*).

We also found little evidence of spatial heterogeneity in seroconversion. Although the locations of health centres reporting cases in the early stages of the outbreak suggested infection spread outwards from central Suva, we found evidence of DENV infection in all study clusters. This suggests that spatial structure may be more important in driving transmission dynamics early in the outbreak, but might not influence the final attack rate. One limitation of this comparison is that we did not have information on outbreak dynamics in the community: in the surveillance data, we only had the

**Table 5.** Comparison of model performance using AIC and DIC.

| Model | Serological data | AIC | ΔAIC | DIC | ΔDIC |
|---|---|---|---|---|---|
| SEIR | MIA | 716.9 | 66.69 | 625.6 | 35.62 |
| SEIR + climate | MIA | 672.9 | 22.7 | 616.6 | 26.65 |
| SEIR + climate + control | MIA | 650.2 | 0 | 589.9 | 0 |
| SEIR | ELISA | 675.1 | 25.74 | 1219 | 643.2 |
| SEIR + climate | ELISA | 668.4 | 19.09 | 599.3 | 23.52 |
| SEIR + climate + control | ELISA | 649.3 | 0 | 575.8 | 0 |

DOI: https://doi.org/10.7554/eLife.34848.019

**Table 6.** Parameter estimates for the 2013/14 dengue epidemic when the model was fitted to MIA or ELISA data.

Median estimates are shown, with 95% credible intervals shown in parentheses. Mean $R_0$ is the average basic reproduction number over a year. Proportion reported was calculated by sampling from the negative binomial distribution that defines the model observation process (i.e. the credible interval reflects both underreporting and dispersion in weekly case reporting). $I_{hc}^0$ and $I_{ha}^0$ denote the number of initially infectious individuals in the younger and older age group respectively.

| Parameter | MIA | ELISA |
| --- | --- | --- |
| Mean $R_0$ | 1.12 (1.02–1.25) | 1.49 (1.35–1.69) |
| Peak $R_0$ | 1.87 (1.7–2.07) | 2.5 (2.29–2.81) |
| Control reduction | 0.57 (0.42–0.82) | 0.70 (0.37–0.95) |
| Proportion reported, lab (%) | 11 (1.1–39) | 13 (2.6–36) |
| Proportion reported, DLI (%) | 9.3 (0.99–37) | 12 (2.8–35) |
| $I_{hc}^0$ | 140 (18–550) | 0.98 (0.21–3.8) |
| $I_{ha}^0$ | 130 (19–680) | 1.3 (0.0094–57) |

DOI: https://doi.org/10.7554/eLife.34848.020

location of the health centres that cases reported to, rather than the location where infection likely occurred.

Analysis of risk factors suggested that presence of self-reported symptoms between 2013–15 was associated with DENV infection. There was also a strong association between rise in ELISA value and self-reported symptoms in individuals who were likely infected, which suggests that raw values from serological tests could potentially be used to estimate the proportion of a population who were asymptomatic during a dengue outbreak, even in older age groups that were already seropositive. However, it is worth noting that the questionnaire that accompanied the serosurvey was brief and only asked about fever and visits to a doctor with fever; there may be specific factors that can better predict prior infection in such settings. We also conducted the follow up survey around 18 months after the outbreak, which means recall bias is a potential limitation of the risk factor analysis. We did not identify environmental factors that were significantly associated with infection, likely as a result of the relatively small sample size in the serological survey, but the estimated odds ratios were broadly consistent with factors that would be expected to increase or decrease infection risk (*Table 3*).

To investigate potential explanations for the outbreak decline in early 2014, we fitted a transmission dynamic model with two human age groups to both surveillance and serological survey data. Our analysis shows the benefits of combining multiple data sources: with surveillance data alone, it would not have been possible to distinguish between self-limiting outbreak driven by a decline in the susceptible population, and one that had ceased for another reason. With the addition of serological data in the model fitting, however, our model was able to quantify the relative contribution of herd immunity, climate and control measures to the outbreak dynamics. In particular, this model suggested that seasonal variation in transmission and herd immunity alone could not explain the fall in transmission. However, an additional decline in transmission in March 2014, which coincided with a nationwide vector clean-up campaign, could better capture the observed patterns in serological and surveillance data.

There are some limitations to our modelling analysis. First, we assumed that seropositivity in IgG antibody tests was equivalent to protective immunity. High levels of neutralising antibodies have been shown to correlate with protection from symptomatic infection (*Katzelnick et al., 2016*), but it remains unclear precisely how much an individual with a given ELISA or MIA value contributes to transmission. Second, we focused on seroprevalence against DENV-3 in the main modelling analysis. As prior infection with one dengue serotype can lead to a cross-reactive immune response against other serotypes (*Guzmán and Kourí, 2002*), we fitted the model to ELISA results (which were not serotype specific) as a sensitivity analysis; this produced the same overall conclusions about which model performed best. Third, we used a flexible time-dependent transmission rate to capture a

potential reduction in transmission as a result of control measures in March 2014. The clean-up campaign included multiple concurrent interventions, which occurred alongside ongoing media coverage of the outbreak; it was therefore not possible to untangle how specific actions – such as vector habitat removal or changes in community behaviour that reduced chances of being bitten – contributed to the outbreak decline. Moreover, factors unrelated to control, such as spatial structure or local weather effects, may also have contributed to the observed decline in transmission; there was heavy rain and flooding in Viti Levu at the end of February 2014 (*ABC News, 2014*).

Although we used a simple function to capture the potential impact of rainfall on vector density, it is unlikely that a more detailed mechanistic relationship would improve the model fit. The peak in rainfall in 2013/14 coincided with the peak in dengue cases; for rainfall to have strongly influenced observed transmission via a reduction in larval carrying capacity, it would need to have peaked earlier, to account for the time delays involved in the vector life cycle (*Lourenço et al., 2017*). We also assumed that all of the population could potentially be infected in the model. Some of the discrepancy between the high attack rate predicted by a randomly mixing model and lower observed seroconversion could in theory be explained by heterogeneity in transmission (*Funk et al., 2016*), which would be expected to reduce the overall proportion infected during an outbreak. If such heterogeneity exists, it is unlikely to act in an 'all-or-nothing' manner over time, with the same individuals remaining at low risk: the high level of seroprevalence in older age groups suggests that only a small proportion of individuals have consistently avoided infection (*Figure 3*).

Finally, our analysis focused on Central Division, Fiji. However, much of the data used in our model – such as surveillance data, post-outbreak serology, and climate information – would be available for other settings. For factors that are harder to measure without paired serology, like age-specific infection rates and potential effectiveness of control measures, a joint inference approach could be employed that combines prior distributions based on the data presented here with available outbreak data from the other location of interest (*Funk et al., 2016*).

Despite these caveats, our results show that transmission dynamic models developed using a combination of serological surveys and surveillance data can be valuable tool for examining dengue fever outbreaks. As well as providing insights into the transmission and control of dengue, the analysis has implications for forecasting of future epidemics. During February and March 2014, members of the research team based at London School of Hygiene and Tropical Medicine provided real-time analysis and outbreak projections for the Fiji National Centre for Communicable Disease Control, to support public health planning (*Nand et al., 2016*). However, a lack of serological data at the time meant it was necessary to make strong assumptions about pre-existing population immunity. With up-to-date population representative serology now available, forecasting models during future outbreaks will be able to include a more realistic herd immunity profile from the outset. Such seroepidemiological approaches could also be employed in other settings, to provide improved forecasts of dengue transmission dynamics and potential disease burden prior to and during outbreaks, as well as quantitative retrospective evaluation of the effectiveness of control measures.

## Materials and methods

### Surveillance data

In December 2013, the dengue outbreak in Fiji was determined to be due to DENV-3 by RT-PCR performed on serum samples sent to the World Health Organization Collaborating Centre for Arbovirus Reference and Research at the Queensland University of Technology (QUT, Brisbane). Hereafter, samples that were ELISA reactive for NS1 antigen or IgM were presumed to be to DENV-3 infections with a sub-sample of them sent for confirmatory serotyping at QUT, the Institut Louis Malardé (ILM) and the US Centers for Disease Control and Prevention. Of the 10,442 laboratory tested cases that were notified to the Fiji National Centre for Communicable Disease Control between 27th October 2013 and 4th March 2014, 4115 (39.4%) were reactive for DENV NS1 and/or anti-DENV IgM. After this time period, dengue surveillance was transitioned from laboratory to clinical-based reporting (i.e. dengue-like illness, DLI) due to the size of the outbreak (*Figure 6*).

Between 27th October 2013 and 31st August 2014, 25,494 suspected cases of dengue (i.e. laboratory tested or confirmed or DLI) were notified to the Fijian Ministry of Health. Of these, 12,413 (48.7%) cases were in Central Division, predominantly in the greater Suva area (*Figure 1*). 10,679

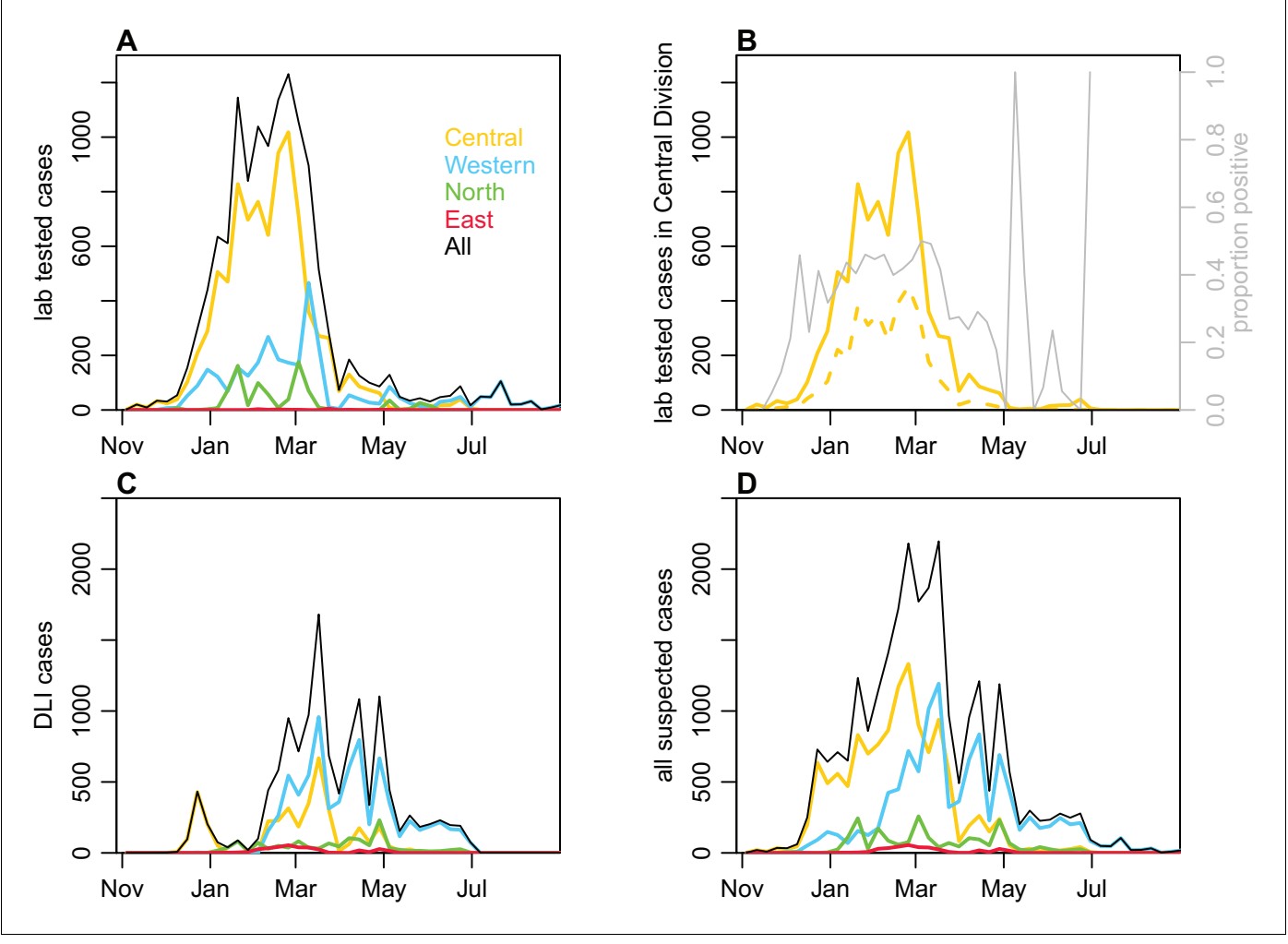

**Figure 6.** Geographical distribution of cases reported each week. (**A**) Lab -tested dengue cases reported in Northern (green), Western (blue) and Central (yellow) divisions between 27th October 2013 and 31st August 2014. (**B**) Total tested and confirmed cases in Central division (solid and dashed lines respectively), as well as proportion of cases that tested positive (grey line). (**C**) Dengue-like illness (DLI) over time. (**D**) Total suspected cases (i.e. tested and DLI).

DOI: https://doi.org/10.7554/eLife.34848.021

The following figure supplements are available for figure 6:

**Figure supplement 1.** Dates of serological sample collection.

DOI: https://doi.org/10.7554/eLife.34848.022

**Figure supplement 2.** Distribution of individual-level measured responses in (**A**) neutralisation assay and (**B**) MIA.

DOI: https://doi.org/10.7554/eLife.34848.023

cases were reported in the Western Division, 2048 cases were reported in the Northern division, largely in or near Labasa, the largest town of Vanua Levu island, and 354 cases were reported in the Eastern Division. For the lab-confirmed cases, date of testing was used to compile weekly case incidence time series; for the DLI data, date of presentation to a health centre was used, as these dates were most complete. Filter paper-based surveillance conducted by ILM between December 2013 and October 2014 found 24 samples positive for DENV-3 by RT-PCR, as well as three samples positive for DENV-2 and one for DENV-1. During 2014/15, there was a flare up of DENV-2 in Fiji. However, relatively few cases occurred on Viti Levu: of the 543 confirmed cases nationally between 1st January 2015 and 29th April 2015, 437 cases (80%) were from the Northern Division (**World Health Organisation, 2015**).

## Serological survey

We conducted a serological survey using pre- and post-outbreak sera from 23 communities in Central Division. Pre-outbreak sera were collected as part of population representative community-based surveys of leptospirosis and typhoid conducted in Central Division between September and November 2013 (*Lau et al., 2016*; *Watson et al., 2017*). Population-proportionate sampling was used to select local nursing zones (the smallest administrative unit). From each of these zones, one community was randomly selected, followed by 25 households from each community and one individual from each of the households. Coincidentally, the sample collection in Central Division finished the same week as the first dengue cases were reported (*Figure 6—figure supplement 1*). Post-outbreak sera were collected during a follow-up study carried out in October and November 2015. Field teams visited participants in Central Division who had previously participated in the 2013 serological study and had consented to being contacted again for health research.

Participants who gave informed consent for the 2015 study completed a questionnaire and provided a 5 ml blood sample. The study was powered to measure the rise in prevalence of anti-DENV antibodies between 2013–15. Historical dengue outbreaks in Fiji (*Table 1*) suggested we would expect to see seroconversion in at least 20% of the study population. Allowing for 5% seroreversion, and 0.05 probability of type-1 error, McNemar's test suggested 250 paired samples could detect a 15% increase in seroprevalence with 95% power, and a 20% increase with ≈100% power. We also collected data on potential risk factors and healthcare-seeking behaviour during this period. The questionnaire asked for details of fever and related visits to a doctor in the preceding two years, and the same for household members in the preceding two years. The questionnaire also recorded details of household environment, including potential mosquito breeding grounds (*Supplementary file 2*).

## Ethical considerations

The 2013 typhoid and leptospirosis studies and the 2015 follow up study were approved by the Fiji National Research Ethics Review Committee (ref 2013–03 and 2015.111.C.D) and the London School of Hygiene and Tropical Medicine Observational Research Ethics Committee (ref 6344 and 10207). Participants in the 2015 follow up study were people who had previously given informed consent to have their blood tested as part of a public health serum bank established in the 2013 typhoid and leptospirosis serosurvey, and agreed to be contacted again by public health researchers. The study was explained in English or the local iTaukei language by bilingual field officers, at the potential participants' preference. Adults gave written informed consent, or thumbprinted informed consent witnessed by a literate adult independent from the study. For children age 12–17 years, written consent was obtained from both the parent and the child. For children aged under 12 years, written consent was obtained from the parent only, though information was provided to both.

## Serological testing of paired sera

Paired pre- and post-outbreak serum samples were tested using an indirect IgG ELISA kit (PanBio Cat No 01PE30), according to manufacturer guidelines. This assay employs recombinant DENV envelope proteins of all four serotypes (*McBride et al., 1998*). Samples with ELISA value of ≤9 PanBio units were defined as seronegative, ≥11 PanBio units seropositive, and values between 9 and 11 as equivocal. Seroconversion was defined as a change from seronegative to seropositive status. Because the indirect IgG ELISA does not distinguish between DENV serotypes, samples were also tested against each of the four specific DENV serotypes using a recombinant antigen-based microsphere immunoassay (MIA), as previously used to examine seroprevalence against different flaviviruses in French Polynesia (*Aubry et al., 2017*, *2018*). Specifically, we wanted to measure the change in seropositivity to DENV-3 during the study period. As an additional validation, a subset of fifty samples from Central Division – including a mixture of those seronegative and seropositive by ELISA and MIA – were tested for the presence of neutralising antibodies against each of the four DENV serotypes using a neutralisation assay as previously described (*Cao-Lormeau et al., 2016*). A neutralisation titre of ≥20 was defined as seropositive (*Figure 6—figure supplement 2A*). For both MIA and neutralisation assay results, the largest change in seropositivity was for DENV-3 (*Supplementary file 1B*). When seropositivity to any DENV (i.e. seropositive to at least one serotype)

was compared, a similar change was observed across ELISA, MIA and neutralisation assay results between 2013 and 2015.

## Serological modelling

Based on ELISA seropositivity in 2015 alone, it would not be possible to identify infections during the 2013/14 outbreak among individuals who were initially seropositive in 2013. We therefore examined the changes in paired individual-level ELISA values between 2013 and 2015. To estimate the probability that a given increase in ELISA value was the result of a genuine rise rather than measurement error, we fitted a two distribution mixture model to the distribution of changes in value between 2013 and 2015. We used a normal distribution with mean equal to zero to reflect measurement error, and a gamma distribution to capture a rise that could not be explained by the symmetric error function. The observed changes in ELISA value we fitted to ranged from −6 to 20; we omitted two outliers that had a change in value of −9 between 2013 and 2015, as these could not be explained with a normally distributed measurement error function. It was not possible to perform the same analysis using the MIA data because unlike the ELISA and neutralisation assay data, the raw MIA values did not follow a bimodal distribution that indicated likely naive and previously exposed individuals (*Figure 6—figure supplement 2B*). We used a generalized additive model with binomial link function to examine the relationship between ELISA value in 2013 and rise between 2013 and 2015, with data points weighted by probability that the change in ELISA value was the result of a genuine rise rather than measurement error. Risk factor analysis was performed using a univariable logistic regression model. Both were implemented using the mgcv package in R version 3.3.1 (*Wood, 2006*; *R Core Team, 2015*).

## Transmission model

### Model structure

We modelled DENV transmission dynamics using an age-structured deterministic compartmental model for human and vector populations, with transitions between compartments following a susceptible-exposed-infective-removed (SEIR) structure (*Kucharski et al., 2016*; *Manore et al., 2014*; *Pandey et al., 2013*). As human population size was known, but the vector population was not, the human compartments were specified in terms of numbers and vectors in terms of proportions. Upon exposure to infection, initially susceptible humans ($S_h$) transitioned to a latent class ($E_h$), then an infectious class ($I_h$) and finally a recovered and immune class ($R_h$). The mosquito population was divided into three classes: susceptible ($S_v$), latent ($E_v$), and infectious ($I_v$). Mosquitoes were assumed to be infectious until they died. We had two human age groups in the model: aged under 20 (denoted with subscript $c$), and aged 20 and over (denoted with subscript $a$). We included births and deaths for the vector population, but omitted human births and deaths because the mean human lifespan is much longer than the duration of the outbreak. The model was as follows:

$$dS_{hc}/dt = -\beta_h(t)S_{hc}I_v \tag{1}$$

$$dE_{hc}/dt = \beta_h(t)S_{hc}I_v - \nu_h E_{hc} \tag{2}$$

$$dI_{hc}/dt = \nu_h E_{hc} - \gamma I_{hc} \tag{3}$$

$$dR_{hc}/dt = \gamma I_{hc} \tag{4}$$

$$dS_{ha}/dt = -\beta_h(t)S_{ha}I_v \tag{5}$$

$$dE_{ha}/dt = \beta_h(t)S_{ha}I_v - \nu_h E_{ha} \tag{6}$$

$$dI_{ha}/dt = \nu_h E_{ha} - \gamma I_{ha} \tag{7}$$

$$dR_{ha}/dt = \gamma I_{ha} \tag{8}$$

$$dC/dt = \nu_h(E_{hc} + E_{ha}) \tag{9}$$

$$dS_v/dt = \delta(t) - \beta_v(t)S_v\left(\frac{I_{hc} + I_{ha}}{N}\right) - \delta(t)S_v \tag{10}$$

$$dE_v/dt = \beta_v(t)S_v\left(\frac{I_{hc} + I_{ha}}{N}\right) - \nu_v(t)E_v - \delta(t)E_v \tag{11}$$

$$dI_v/dt = \nu_v(t)E_v - \delta(t)I_v \tag{12}$$

The compartment $C$ recorded the cumulative total number of human infections, which was used for model fitting. Based on most recent Fiji census in 2007, we set the population size $N$ to be 342,000 in Central Division (*Fiji Bureau of Statistics, 2007*), and split this population between the two age groups based on the populations of each reported in the census ($N_c$=133,020 and $N_a$=208,980). We estimated two initial conditions for each human age group: the initial number of infective individuals, $I_h^0$, and the initial number immune, $S_h^0$. We assumed that there were the same number of individuals initially exposed as there are individuals infectious (i.e. $E_h^0 = I_h^0$). For the vector population, we only estimated the initial proportion infectious. We assumed that $E_v^0 = I_v^0$ and the remaining proportion of mosquitoes were susceptible. We assumed that the mean intrinsic latent period, $1/\nu_h$, and human infectious period, $1/\gamma$ remained constant over time, with informative priors (*Table 7*). As detailed in the sections below, the following parameters were time dependent: transmission rate from vectors to humans, $\beta_h(t)$; transmission rate from humans to vectors, $\beta_v(t)$; mosquito lifespan, $1/\delta(t)$; and extrinsic latent period, $1/\nu_v(t)$. To avoid infection declining to implausibly

**Table 7.** Parameters fitted in the model.
Prior distributions are given for all parameters, along with source if the prior incorporates a specific mean value. All rates are given in units of days$^{-1}$.

| Parameter | Definition | Prior | Source |
|---|---|---|---|
| $1/\nu_h$ | intrinsic latent period | Gamma($\mu$=5.9, $\sigma$=0.1) | (*Chan and Johansson, 2012*) |
| $1/\gamma$ | human infectious period | Gamma($\mu$=5, $\sigma$=0.1) | (*Duong et al., 2015*) |
| $1/\hat{\nu}_v$ | extrinsic latent period at 25° | Gamma($\mu$=10, $\sigma$=0.1) | (*Mordecai et al., 2017*; *Chan and Johansson, 2012*) |
| $1/\hat{\delta}$ | mosquito lifespan at 25° | Gamma($\mu$=8, $\sigma$=0.1) | (*Sheppard et al., 1969*) |
| $\hat{\alpha}$ | biting rate at 25° | Gamma($\mu$=0.25, $\sigma$=0.1) | (*Mordecai et al., 2017*) |
| $\hat{m}$ | baseline vector density | $\log\mathcal{U}(0, 20)$ | (*Andraud et al., 2012*) |
| $\hat{K}$ | carrying capacity scaling parameter | $\log\mathcal{U}(0, 100)$ | |
| $a_1$ | gradient of sigmoidal change in transmission | $\log\mathcal{U}(0, 1000)$ | |
| $a_2$ | magnitude of sigmoidal change in transmission | $\log\mathcal{U}(0, 1)$ | |
| $a_\tau$ | timing of sigmoidal change in transmission | $\log\mathcal{U}$(8th March 2014, 5th April 2014) | (*Break Dengue, 2014*) |
| $r_{lab}$ | proportion of cases reported as lab tested | $\log\mathcal{U}(0, 1)$ | |
| $r_{DLI}$ | proportion of cases reported as DLI | $\log\mathcal{U}(0, 1)$ | |
| $\rho$ | reporting dispersion | $\log\mathcal{U}(0, \infty)$ | |
| $I_{hc}^0$ | initial number infectious aged < 20 | $\log\mathcal{U}(0, N_c)$ | |
| $R_{hc}^0$ | initial number immune aged < 20 | $\log\mathcal{U}(0, N_c)$ | |
| $I_{ha}^0$ | initial number infectious aged 20+ | $\log\mathcal{U}(0, N_a)$ | |
| $R_{ha}^0$ | initial number immune aged 20+ | $\log\mathcal{U}(0, N_a)$ | |
| $I_v^0$ | initial proportion of infectious mosquitoes | $\log\mathcal{U}(0, 1)$ | |

DOI: https://doi.org/10.7554/eLife.34848.024

small levels then rising again in the following season, we included potential for extinction in the model. If the number of individuals in any of the $E$ or $I$ human compartments dropped below one, the model set the value to zero. Hence if there were no exposed or infectious individuals in either of the age groups, the epidemic would end.

## Seasonal parameter variation

We assumed that the vector-specific parameters varied over time in the model, as a result of seasonal changes in temperature and rainfall (*Descloux et al., 2012*). During 2013/14 in Central Division, average monthly rainfall ranged from around 100 to 400 mm, and daily temperature varied between 21 and 30°C (*The World Bank, 2016*; *Fiji Meteorological Service, 2017*). Temperature reached its maximum in January/February, and minimum in August/September (*Figure 5—figure supplement 1A*). As the daily temperature data were noisy and surveillance data were only available on a weekly timescale, in the model we defined $\text{temp}_t$ as the seven day moving average of temperature on day $t$ (i.e. the average temperature over the preceding week). We also defined $\text{rain}_t$ as the average rainfall on day $t$, interpolated from monthly data (*Figure 5—figure supplement 1B*).

Based on estimated mechanistic relationships between temperature and *Aedes aegypti* dynamics (*Lourenço et al., 2017*; *Mordecai et al., 2017*), we assumed that the following vector-specific parameters were temperature dependent: extrinsic incubation period, $1/\nu_v(t)$; lifespan, $1/\delta(t)$; biting rate, $\alpha(t)$; probability of transmission to a human, $p_{vh}(t)$; and probability of infection from an infectious human, $p_{hv}(t)$. We incorporated these temperature-dependent dynamics using symmetric ($\phi$) and asymmetric ($\psi$) unimodal thermal response functions (*Mordecai et al., 2017*; *Briere et al., 1999*):

$$\phi(x, y, T_m, T_0) = \begin{cases} \min\{y(x - T_0)(T_m - x), 1\} & \text{if } T_0 < x < T_m \\ 0 & \text{else} \end{cases} \tag{13}$$

$$\psi(x, y, T_m, T_0) = \begin{cases} \min\{yx(x - T_0)\sqrt{T_m - x}, 1\} & \text{if } T_0 < x < T_m \\ 0 & \text{else} \end{cases} \tag{14}$$

The parameters were defined using the median estimated value from these functions fitted to empirical data (*Mordecai et al., 2017*):

$$1/\nu_v(t) = 1/(\hat{\nu}_v \phi(\text{temp}_t, 6.11 \times 10^{-5}, 45.53, 10.30)/0.10) \tag{15}$$

$$1/\delta(t) = 1/(\hat{\delta} \psi(\text{temp}_t, 9.02, 37.66, -0.14)/29.00) \tag{16}$$

$$\alpha(t) = \hat{\alpha} \phi(\text{temp}_t, 0.00020, 40.04, 13.76)/0.22 \tag{17}$$

$$p_{vh}(t) = \phi(\text{temp}_t, 0.00083, 35.78, 17.23) \tag{18}$$

$$p_{hv}(t) = \phi(\text{temp}_t, 0.00049, 37.38, 12.67) \tag{19}$$

Here $1/\nu_v(t)$, $1/\delta(t)$ and $\alpha(t)$ are normalised so that they equal $1/\hat{\nu}_v$, $1/\hat{\delta}$, and $\hat{\alpha}$ respectively when $\text{temp}_t = 25$°C. In the model, most of these parameters varied monotonically within the temperature range observed in Fiji (*Figure 5—figure supplement 1C–G*). We used informative priors for the average extrinsic latent period, $1/\hat{\nu}_v$, mosquito lifespan, $1/\hat{\delta}_v$, and biting rate, $\hat{\alpha}$ (*Table 7*).

We assumed that vector density, $m(t)$, could vary with both temperature and rainfall (*Figure 5—figure supplement 1H–I*). The contribution of vector density to transmission was influenced by four factors (*Mordecai et al., 2017*): fecundity, $f$ (i.e. number of eggs produced per female mosquito per day); egg-to-adult survival probability, $e$, the mosquito development rate, $d$, and the larval carrying capacity $K$. In the model, vector density over time was equal to:

$$m(t) = m(\text{temp}_t, \text{rain}_t) \tag{20}$$

$$= \frac{\hat{m}}{m_0} e(t) f(t) d(t) \frac{K(t)}{1 + K(t)} \tag{21}$$

where $\frac{\hat{m}}{m_0}$ is a scaling term and the $K/(1+K)$ term incorporating carrying capacity follows from the equilibrium solution of the logistic growth model (*Pearl and Reed, 1920*) (*Figure 5—figure supplement 1I*). We assumed that $d$, $e$, and $f$ were temperature dependent, based on functions fitted to empirical data (*Mordecai et al., 2017*), and $K$ was linearly dependent on rainfall:

$$d(t) = \phi(\text{temp}_t, 7.84e \times 10^{-5}, 39.10, 11.56) \tag{22}$$

$$e(t) = \psi(\text{temp}_t, 13.58, 38.29, -0.0060) \tag{23}$$

$$f(t) = \phi(\text{temp}_t, 0.0082, 34.44, 14.78) \tag{24}$$

$$K(t) = \hat{K} \text{rain}_t / 222.44 \tag{25}$$

$K(t)$ was normalised so its mean value over the year was equal to $\hat{K}$ and we set $m_0 = 0.5752381$ so that $m(t) = \hat{m}K/(1+K)$ when the temperature was 25°. Prior distributions for parameter values are given in *Table 7*. In the absence of control measures, the vector-to-human, $\beta_h$, and human-to-vector, $\beta_v$, transmission rates were therefore:

$$\beta_v(t) = \alpha(t) p_{vh}(t) \tag{26}$$

$$\beta_h(t) = \alpha(t) p_{hv}(t) m(t) \tag{27}$$

## Control measures

To capture the potential additional reduction in transmission over time as a result of the national clean-up campaign between 8th and 22nd March 2014, we used a flexible sigmoid function:

$$\chi(t) = \left( 1 - \frac{a_2}{1 + e^{-a_1(t - a_\tau)}} \right) \tag{28}$$

We constrained this function so that the midpoint, $a_\tau$, was between the start date of the campaign, 8th March 2014, and 5th April 2014, four weeks later (*Figure 5—figure supplement 1J*). We assumed that this function acted to reduce the vector-to-human transmission rate:

$$\beta_h(t) = \alpha(t) m(t) \chi(t) \tag{29}$$

There were multiple concurrent interventions during the clean-up campaign, including promotion of awareness about protection from bites as well as larval habitat removal. Given the structure of the data available, it would not be possible to independently estimate the extent to which the campaign directly reduced vector-to-human transmission, that is $\chi(t)$ acting on $\alpha(t)$, rather than vector density, that is $\chi(t)$ acting on $m(t)$. However, if there had been a substantial effect on larval habitat capacity but not on biting rate, we may expect to infer a larger value of $a_\tau$, to reflect the delay in impact as a result of the time required for vector development.

## Effective reproduction number

The next generation matrix for humans and vectors was defined as follows (*Manore et al., 2014, 2017*):

$$\begin{pmatrix} R_{hh} & R_{hv} \\ R_{vh} & R_{vv} \end{pmatrix} = \begin{pmatrix} 0 & \frac{\beta_h(S_{hc} + S_{ha})\nu_v}{\delta(\delta + \nu_v)N} \\ \frac{\beta_v S_v}{\gamma} & 0 \end{pmatrix}$$

and the effective reproduction number, $R$, was equal to the dominant eigenvalue of this matrix. The

basic reproduction, $R_0$, was calculated by the same method, but assuming that both humans and vectors were fully susceptible.

## Model fitting

The model was jointly fitted to laboratory-confirmed case data and serological data using Markov chain Monte Carlo (MCMC) via a Metropolis-Hastings algorithm. For the case data, we considered time units of one week. To construct a likelihood for the observed cases, we defined case count for week $t$ as $c_t = C_t - C_{t-1}$.

Because reporting switched from lab tested to DLI during the outbreak, we jointly fitted two sets of time series data. The first dataset was lab tested cases. We defined the first observation as 4th November 2013, the week of the first confirmed case in Central Division, and the last observation as 26th May. The second dataset was DLI cases, which we fitted from 1st February until 26th May. Earlier DLI cases were not included as these were likely to reflect reporting artefacts rather than genuine infections. The two time series we fitted were disjoint: cases were either reported as lab tested or DLI.

We assumed that the two set of observed cases followed a negative binomial distributions with mean $q_t r_{lab} c_t$ and $(1 - q_t) r_{DLI} c_t$ respectively, and a shared dispersion parameter $\rho$, to account for potential temporal variability in reporting (*Bretó et al., 2009*). We used a negative binomial distribution to allow for both under- or over-reporting, the latter being potentially relevant in the final stages of the outbreak when case numbers were low. Here $q_t$ denotes the proportion of cases in week $t$ that are lab tested rather than reported as DLI. As it was not possible to fit this value for each week, it was fixed in the model as $q_t = y_{lab}/(y_{lab}(t) + y_{DLI}(t))$, where $y_{lab}(t)$ and $y_{DLI}(t)$ are the number of observed lab tested and DLI cases in week $t$ respectively. The total expected number of reported cases in week $t$ was therefore equal to $(q_t r_{lab} + (1 - q_t) r_{DLI}) c_t$.

As well as fitting to surveillance data, we fitted the model to the proportion of each age group immune (as measured by seroprevalence) at the start and end of the outbreak. Let $X_{ij}$ be a binomially distributed random variable with size equal to the sample size in group $i$ and probability equal to the model predicted immunity in year $j$, and $z_{ij}$ be the observed seroprevalence in group $i$ in year $j$. The overall log-likelihood for parameter set $\theta$ given case data $Y = \{y_t\}_{t=1}^T$ and serological data $Z = \{z_{ij}\}_{i \in \{1,2\}, j \in \{2013,2015\}}$ was therefore:

$$L(\theta \sim | \sim Y, Z) = \sum_t \log P(y_t | c_t) + \sum_{i=1}^{2} \sum_{j \in \{2013,2015\}} \log P(X_{ij} = z_{ij}) \tag{30}$$

We considered four model scenarios: an SEIR model without climate-driven variation or control, fitted to surveillance data only; SEIR model without climate-driven variation or control, fitted to surveillance and serological data; SEIR model with climate-driven variation only, fitted to surveillance and serological data; SEIR model with climate-driven variation and control, fitted to surveillance and serological data. We fitted the model to either MIA or ELISA data, to reflect two different assumptions about the relationship between seroprevalence and immunity. The model using MIA data made the assumption that only individuals who were seropositive to DENV-3 were immune to this serotype. As a sensitivity analysis, the model using ELISA data assumed that seropositivity to any DENV serotype indicated immunity to DENV-3.

All observations were given equal weight in the model fitting. The joint posterior distribution of the parameter set $\theta$ was obtained from 200,000 MCMC iterations, each with a burn-in period of 20,000 iterations. We used adaptive MCMC to improve efficiency of mixing: we iteratively adjusted the magnitude of the covariance matrix used to resample $\theta$ to obtain a target acceptance rate of 0.234 (*Roberts and Rosenthal, 2009*). Posterior estimates for MIA and ELISA data are shown in *Supplementary files 1C–D*. The statistical and mathematical models were implemented in R version 3.3.1 (*R Core Team, 2015*) using the deSolve package (*Soetaert et al., 2010*) and parallelised using the doMC library (*Revolution Analytics, 2014*).

## Model comparison

We compared the performance of different models using the deviance information criterion (DIC), which accounts for the trade off between model fit and complexity (*Spiegelhalter et al., 2002*). The

deviance of a model for a given parameter set, $\theta$, is given by $D(\theta) = -2L(\theta|Y,Z)$. The DIC is therefore:

$$DIC = D(\bar{\theta}) + \mathrm{var}(D(\theta)) \tag{31}$$

where $\bar{\theta}$ is the median of $\theta$ with respect to the posterior distribution and $\mathrm{var}(D(\theta))/2$ is the effective number of parameters. The median of $\theta$ as used rather than mean because the likelihood was non-log-concave in $\theta$, which meant that the posterior mean was a poor estimator (*Spiegelhalter et al., 2002*). As an additional validation, we compared models using the Akaike information criterion (AIC), which accounts for the trade off between model fit and complexity (*Akaike, 1973*). The AIC of a model for a given parameter set, $\theta$, is given by $AIC = -2\hat{L}(\theta) + 2n_{param}$ where $\hat{L}(\theta)$ is the maximised value of the likelihood and $n_{param}$ is the number of parameters.

## Data availability

Serological, surveillance and climate data are provided in *Supplementary file 3*. Code and data required to reproduce the main serological and modelling analysis are available at: https://github.com/adamkucharski/fiji-denv3-2014. Copy archived at https://github.com/elifesciences-publications//fiji-denv3-2014.

## Acknowledgements

We warmly thank all the participants and community leaders who generously contributed to the study. We are also grateful to Kylie Jenkins of Australian Aid's Fiji Health Sector Support Programme, Teheipuaura Mariteragi-Helle at the Institut Louis Malardé, and Dr Ketan Christie at the University of the South Pacific. We thank the staff of the Ministry of Health clinical services and Mataika House Fiji Centre for Communicable Disease Control for providing the surveillance data underpinning this study. We would also like to acknowledge the work of the field teams: Dr. Kitione Rawalai, Jeremaia Coriakula, Ilai Koro, Sala Ratulevu, Ala Salesi, Meredani Taufa, and Leone Vunileba (2013); Meredani Taufa, Adi Kuini Kadi, Jokaveti Vubaya, Colin Michel, Mereani Koroi, Atu Vesikula, and Josateki Raibevu (2015).

## Additional information

### Competing interests

Stéphane Hué: SH acknowledges the financial contribution of Janssen Sciences Ireland UC towards the research and completion of this work. Martin L Hibberd: MH acknowledges the financial contribution of Janssen Sciences Ireland UC towards the research and completion of this work. The other authors declare that no competing interests exist.

### Funding

| Funder | Grant reference number | Author |
|---|---|---|
| Medical Research Council | MR/K021524/1 | Adam J Kucharski |
| Wellcome Trust | 206250/Z/17/Z | Adam J Kucharski |
| Royal Society | 206250/Z/17/Z | Adam J Kucharski |
| Embassy of France in the Republic of Fiji, Kiribati, Nauru, Tonga and Tuvalu | | Mike Kama |
| French Ministry for Europe and Foreign Affairs | Pacific Funds N°12115-02/09/15 | Mike Kama Van-Mai Cao-Lormeau |
| French Ministry for Europe and Foreign Affairs | Pacific Funds N°03016-20/05/16 | Mike Kama Van-Mai Cao-Lormeau |
| Medical Research Council | MR/J003999/1 | Conall H Watson |

| Commissariat Général à l'Investissement | ANR-10-LABX-62-IBEID | Jessica Vanhomwegen Jean-Claude Manuguerra |
| National Health and Medical Research Council | 1109035 | Colleen L Lau |
| French Ministry for Europe and Foreign Affairs | Pacific Funds N°06314-09/04/14 | Van-Mai Cao-Lormeau |
| Janssen Research and Development | Janssen Sciences Ireland UC | Stéphane Hué Martin L Hibberd |

The funders had no role in study design, data collection and interpretation, or the decision to submit the work for publication.

## Author contributions
Adam J Kucharski, Conceptualization, Data curation, Formal analysis, Investigation, Visualization, Methodology, Writing—original draft, Writing—review and editing; Mike Kama, Conceptualization, Data curation, Formal analysis, Investigation, Writing—review and editing; Conall H Watson, Conceptualization, Data curation, Formal analysis, Investigation, Writing—original draft; Maite Aubry, Data curation, Formal analysis, Writing—review and editing; Sebastian Funk, Alasdair D Henderson, Oliver J Brady, Formal analysis, Writing—review and editing; Jessica Vanhomwegen, Jean-Claude Manuguerra, Resources, Methodology; Colleen L Lau, Conceptualization, Writing—review and editing; W John Edmunds, Conceptualization, Formal analysis; John Aaskov, Van-Mai Cao-Lormeau, Conceptualization, Data curation, Formal analysis, Writing—review and editing; Eric James Nilles, Stéphane Hué, Martin L Hibberd, Conceptualization, Formal analysis, Writing—review and editing

## Author ORCIDs
Adam J Kucharski (iD) http://orcid.org/0000-0001-8814-9421
Sebastian Funk (iD) http://orcid.org/0000-0002-2842-3406
Jean-Claude Manuguerra (iD) http://orcid.org/0000-0002-5202-6531
Eric James Nilles (iD) http://orcid.org/0000-0001-7044-5257

## Ethics
Human subjects: The 2013 typhoid and leptospirosis studies and the 2015 follow-up study were approved by the Fiji National Research Ethics Review Committee (ref 2013-03 and 2015.111.C.D) and the London School of Hygiene and Tropical Medicine Observational Research Ethics Committee (ref 6344 and 10207). Participants in the 2015 follow-up study were people who had previously given informed consent to have their blood tested as part of a public health serum bank established in the 2013 typhoid and leptospirosis serosurvey, and agreed to be contacted again by public health researchers. The study was explained in English or the local iTaukei language by bilingual field officers, at the potential participants' preference. Adults gave written informed consent, or thumb-printed informed consent witnessed by a literate adult independent from the study. For children aged 12–17 years, written consent was obtained from both the parent and the child. For children aged under 12 years, written consent was obtained from the parent only, though information was provided to both.

## Decision letter and Author response
Decision letter https://doi.org/10.7554/eLife.34848.029
Author response https://doi.org/10.7554/eLife.34848.030

# Additional files
## Supplementary files
• Supplementary file 1. Supplementary data and results tables.
DOI: https://doi.org/10.7554/eLife.34848.025

• Supplementary file 2. Questionnaire that accompanied the 2015 serological survey in Central Division.

DOI: https://doi.org/10.7554/eLife.34848.026

• Supplementary file 3. Serological, surveillance and climate data for the 2013/14 DENV-3 outbreak in Central Division.
DOI: https://doi.org/10.7554/eLife.34848.027

## Data availability

Surveillance and serological data are provided as supporting information. Full code and data required to reproduce the main serological and modelling analyses are available at: https://github.com/adamkucharski/fiji-denv3-2014. The raw GPS data cannot be made publicly available as this contains potentially identifiable information.

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
