## [Decision Letter]

Thank you for submitting your article "Using paired serology and surveillance data to quantify dengue transmission and control during a large outbreak in Fiji" for consideration by *eLife*. Your article has been reviewed by two peer reviewers, one of whom is a member of our Board of Reviewing Editors, and the evaluation has been overseen by Prabhat Jha as the Senior Editor. The reviewers have opted to remain anonymous.

The reviewers have discussed the reviews with one another and the Reviewing Editor has drafted this decision to help you prepare a revised submission.

Summary:

This paper presents a modelling analysis of an island epidemic of dengue-3 in 2013/4 giving insight into the effect of seasonality and control measures on transmission.

Essential revisions:

- Greater sensitivity analysis and caveats regarding assumptions about vector population dynamics (see both reviews), including modelling larval carrying capacity or at least rainfall-driven recruitment rates.

- On a related topic – both reviewers comment on how seasonality was included for some parameters but not others. The main demonstrated effects of temperature are on EIP and mortality, plus rainfall via carrying capacity. There's not much evidence around temperature affecting biting rates/transmission coefficients. This aspect needs to be approached more systematically, given the effects of seasonality and of the clean-up campaign may be confounded.

- More detail on the MIA data is needed – as was done for the ELISA data, raw data should be shown (for control and DENV3 antigens), and arguably a mixture model fitted. More justification of the use of 2 assays is also needed.

- Excluding data from the key weeks around the peak of the epidemic is problematic. Suggestions as to how this might be avoided are given in the detailed reviewer comments.

- The Materials and methods section needs to include detail of the model comparison exercise undertaken, and tables of parameter estimates included in the Supplementary Information.

- All datasets used need to be made public with the paper.

Reviewer #1:

This paper presents a nicely detailed study of an island epidemic of dengue-3 in 2013/4. The most novel aspect of the study is the model-based estimation of the possible impact of a control program implemented during the epidemic. The authors make a reasonable case that there is evidence that this program had a moderate impact on transmission but to strengthen the robustness of this conclusion, aspects of the analysis require clarification and further sensitivity analysis. Detailed comments follow (but should not be viewed in light of my overall positive view):

- The transmission model assumes a constant mosquito population size (Equation 0.10). Did the authors attempt to fit a model with density dependent regulation of larval populations, perhaps where carrying capacity is driven by rainfall: i.e. K= c + m R, where R is accumulated rainfall over some time interval T (e.g. 1 month)? This might allow a better fit to the epidemiological data – and might eliminate any statistically significant impact of the clean-up campaign, given the sharp drop in rainfall in May and the peak in April.

- Why assume sinusoidal variation in temperature (Equation 0.13), given the actual data is available (albeit I accept it looks reasonably sinusoidal). Then the functional form used in Mordecai et al., 2017 could be used directly, with priors on those parameters. This would seem to be more satisfying…

- It would be better to represent the clean-up campaign (which would have reduced larval habitat, not adult mosquito density) as a (perhaps linearly increasing over time) reduction in the mosquito source term (multiplier on recruitment rate δ in Equation 0.10, or of carrying capacity) than a direct modifier of transmissibility. Doing so will offset the impact on transmission by a generation time – i.e. ~2 weeks. It may then be possible to fit a parametrically simpler model to the exact timing of the clean-up campaign.

- Little mention is made of the discordance of the MIA assay with the ELISA results – how was seroconversion measured with that assay? If with just a ratio, then perhaps a mixture model would fit better? It would also be good to see the raw results for that assay – the authors spend a lot of time modelling the ELISA results, but then use the MIA data in their default (best fit) model with no comment on the relative reliability of one vs the other. Figure 3B should also be updated to show the MIA results as well as the ELISA ones. More generally, why were two serological assays used? The motivation for this is never stated.

- Table 2 should have the MIA results for 'Any dengue' shown to be able to compare properly with ELISA, especially given the MIA model is the one presented in Figure 5.

- I found it difficult to reconcile the data plotted in Figure 1A with the data points plotted on Figure 5A. Which region was being fitted to – I presumed the whole country, but if that's the case, I don't know why the case numbers peak at about 600 in Figure 5A but at over 2000 in the black curve of Figure 1A. Are weekly case number being plotted in some places, and monthly in others? If so, I suggest using weekly numbers throughout (and updating figures to state 'weekly cases'). It would also be good if Figure 1A was a bit bigger.

- It was unfortunate that disease surveillance changed almost coincident with the clean-up campaign. But I was uncomfortable with the authors dropping 2-4 weeks of surveillance data from the fit. The authors refer to a sensitivity analysis for the number of weeks of data dropped, but I couldn't find the results. At the very least, cases of the two types for the missing 2 weeks should be plotted in Figure 5A, shown in a different colors.

- I think the authors could avoid dropping those critical weeks of data, Assuming the transition occurred at different times for different health facilities, on week t let p_t_ denote the proportion of surveillance using lab confirmation, and (1-p_t_) be the proportion using DLI. Then the expected total number of reported cases (lab plus DLI) is [p_t_ r_1_ + (1-p) r_2_]c_t_. If O1_t_ and O2_t_ are the reported cases in week t from the two surveillance systems, then O1_t_/O2_t_ gives an estimate of p_t_/(1-p_t_). This chain of reasoning can be represented in the likelihood in a number of ways – by using separate likelihoods for the lab and DLI cases, and estimating p_t_ for each of the 2 currently omitted weeks explicitly, or (more crudely) by pre-calculating p_t_ from O1_t_/O2_t_ for the transition period, and using a single likelihood for total cases (albeit still with r_1_ and r_2_ – though a single k would need to be used in that case). Or there may be some cleverer way to do it! As an aside, I wasn't quite clear why different k values were fitted to the different case types – were the estimates different?

- Wouldn't a beta-binomial model (parameterised in terms of a mean p and a overdispersion parameter) be a better representation of the observation process than a negative binomial?

- Any thoughts on why the first DLI point shown in Figure 5A is so high? Though looking at Figure 1A, perhaps this is an artefact of omitting some data points? Otherwise, could it indicate a transiently negative impact of the intervention? How does the model fit change if that point is excluded (see my early ref to the 2-4 week sensitivity analysis mentioned but not presented)?

- Supplementary File 1 – the EIP is a bit long in my opinion – the blood feeding experiments of Simmons etc. might suggest 10 days is a more reasonable value. Some sensitivity analysis to this would be useful, given changing the generation time will change R_0_ and thus the relationship between infection attack rates and case incidence (perhaps!)

- Regarding Figure S7 and the need for time-varying transmission neglects individual-level heterogeneity in exposure. I get the basic point about the limited serological attack rate, but heterogeneity could in theory explain this. Such heterogeneity is quite extreme for malaria, and likely to be comparable for dengue, given the more limited vector dispersion range. I accept that time-varying transmission is a more likely explanation, but the authors may still want to comment. In addition, while the authors didn't find significant predictors of seroconversion (unsurprising, given the small sample size), the RRs largely agree with intuition.

- The model comparison exercise (Supplementary file 7) should be described in the Materials and methods section.

- Where are all the parameter estimates for the models shown in Table S3? These need to be given. Or at least for the SEIR+climate vs SEIR+climate+control models (ELISA and MIA variants)?

- All data needs to be released with the paper to allow reproducibility of results (i.e. raw data from serosurvey, surveillance data at the resolution used, climate data)

Reviewer #2:

This manuscript combines a mathematical modeling analysis with two empirical data sources pertaining to the 2013-2015 epidemic of dengue in Fiji. One of the most unique aspects of this work is that the authors had the good fortune of having a number of samples collected for other purposes just prior to the epidemic that could be assayed for prior DENV exposure and used to form a longitudinal cohort that was followed up on after the epidemic in 2015. In many cases, epidemic analyses are limited to passive surveillance data, which is also examined here. Thus, this work represents one of a very limited number of opportunities to perform separate and combined analyses with these two different data types, providing insight into the relative strengths and limitations of the two and providing information about the extent of discrepancy between inferences made on one, the other, or both. Moreover, a number of different forms of analysis were conducted, including a mathematical transmission model that was fitted to both data types.

Overall, I view this manuscript as having many strengths and believe that it reflects a nice combination of unique data and thoughtful analysis. My primary criticisms of the paper have to do with where emphasis is placed in terms of results, writing, and take-home messages. My impression from reading this manuscript is that some of the primary results in the authors' view pertain to what factors drove the epidemic and their conclusion that vector control made a perceptible contribution to ending the epidemic. While I concede that these claims are plausible and perhaps even likely, I had reservations about the extent of inference drawn based on the analyses that were performed. The transmission model used here is a standard choice, but the reality is that models of this form were devised for theoretical purposes rather than inferential ones. That is not to say that models of this form cannot be used for inference – indeed, I am engaged in work of that nature myself – but there is a great deal of uncertainty in many structural aspects of these models that must be acknowledged and examined before reliable inferences can be drawn. My concern is that the authors have not done enough in that respect.

To elaborate, in the Abstract, it is stated that "Mathematical modelling showed that temperature-driven variation in transmission and herd immunity could not fully explain observed dynamics." The authors then go on to state that there was an additional reduction in transmission explained by a vector clean-up campaign. While they may be right, a problem with these statements here and elsewhere in the manuscript is that the one relatively simple model chosen was used to make a rather conclusive statement about what did or did not drive the observed dynamics (e.g., "mathematical modeling" rather than "a mathematical model" showed). I can think of numerous ways in which this model could be elaborated on or alternatives proposed (and not necessarily more mechanistic detail but potentially more flexibility from more statistical descriptions in certain places) that would likely better fit the data and could lead to different conclusions. For example, the model did not allow for a dynamic vector population, which could be a major factor in driving seasonal transmission in its own right but also leads to changes in the demographic composition of the vector population that are extremely important epidemiologically and that interact in important ways with control measures. And while there was some allowance for seasonality, it entered the model in a somewhat odd way (effectively influencing biting rate and vector competence but not EIP, mortality, etc.). Given how quickly a list of potentially major concerns about the model's structure can be generated (not to mention the very particular assumption of sinusoidal seasonality, which there is no good reason in principle to expect as opposed to other seasonal forms), it may be overreaching to make strong claims about ruling out or supporting certain factors in driving the epidemic. The results related directly to the model are suggestive, but that's about it.

Inferences based on the serological data are also not completely straightforward. First, there is the difference between what the IgG and MIA assays tell you about: infection with one's first serotype and infection with a particular serotype. Second, the model does not account for multiple serotypes and is thus presumably intended as a DENV-3 model only. The surveillance data are reflective of all DENV serotypes, however, so some discrepancy between models fitted to these data sources is to be expected based on that alone. Whether in fitting the model or performing the age analysis, a lot of care must be taken when dealing with data that speak to non-specific seroconversion. The reason is that there could have been people experiencing their second infection but who did not seroconvert because they were already seroconverted by the time of the period of observation. The MIA data address this issue to a large extent, but that seems to be done in an either/or way using IgG or MIA but not both sources of information. Additionally, I do not have much familiarity with MIA and would have appreciated more exposition about its validity. For example, how do its abilities to infer serotype-specific exposure compare to PRNT? Could some PRNTs be done on samples from this study to demonstrate that? Are results interpretable for individuals with multiple DENV exposures?

---

## [Author Response]

Essential revisions:- Greater sensitivity analysis and caveats regarding assumptions about vector population dynamics (see both reviews), including modelling larval carrying capacity or at least rainfall-driven recruitment rates.

We have updated the manuscript to include rainfall dependency in mosquito density (see response 1.1) as well as discussion of recruitment rates and control measures (see response 1.3).

- On a related topic – both reviewers comment on how seasonality was included for some parameters but not others. The main demonstrated effects of temperature are on EIP and mortality, plus rainfall via carrying capacity. There's not much evidence around temperature affecting biting rates/transmission coefficients. This aspect needs to be approached more systematically, given the effects of seasonality and of the clean-up campaign may be confounded.

We have updated the model to systematically include temperature-driven variation on all relevant vector parameters. The potential confounding of clean-up campaign and seasonality is also discussed in more detail. These changes are detailed in responses 1.1– 1.3 and 2.3–2.4.

- More detail on the MIA data is needed – as was done for the ELISA data, raw data should be shown (for control and DENV3 antigens), and arguably a mixture model fitted. More justification of the use of 2 assays is also needed.

The updated manuscript includes more detail about the MIA data and how it relates to the ELISA results, as well as additional neutralisation assay results for validation (see responses 1.4 and 2.5 below).

- Excluding data from the key weeks around the peak of the epidemic is problematic. Suggestions as to how this might be avoided are given in the detailed reviewer comments.

We have updated the fitting procedure to include both time series during these key weeks (see responses 1.7–1.8 below).

- The Materials and methods section needs to include detail of the model comparison exercise undertaken, and tables of parameter estimates included in the Supplementary Information.

These have now been added to the manuscript, in subsection “Model comparison” and supplementary files showing parameter estimates.

- All datasets used need to be made public with the paper.

We realise we could have been clearer about data availability. The data and code required to reproduce the analysis are already available on GitHub (http://github.com/adamkucharski/fiji-denv3-2014), as noted in the submitted manuscript and transparent reporting form. In the revised manuscript, we have also included a supplementary file with all the relevant data in one place.

Reviewer #1:This paper presents a nicely detailed study of an island epidemic of dengue-3 in 2013/4. The most novel aspect of the study is the model-based estimation of the possible impact of a control program implemented during the epidemic. The authors make a reasonable case that there is evidence that this program had a moderate impact on transmission but to strengthen the robustness of this conclusion, aspects of the analysis require clarification and further sensitivity analysis. Detailed comments follow (but should not be viewed in light of my overall positive view):- The transmission model assumes a constant mosquito population size (Equation 0.10). Did the authors attempt to fit a model with density dependent regulation of larval populations, perhaps where carrying capacity is driven by rainfall: i.e. K=c + m R, where R is accumulated rainfall over some time interval T (e.g. 1 month)? This might allow a better fit to the epidemiological data – and might eliminate any statistically significant impact of the clean-up campaign, given the sharp drop in rainfall in May and the peak in April.

1.1 We did not fit a rainfall-dependent model initially as we noted that the peak in rainfall (April) occurred after the outbreak peak (Feb/Mar), and so it was unlikely that the main decline in transmission was the result of a reduction in carrying capacity. However, we realise that it would be helpful to test this quantitatively and now include a simple function to allow mosquito density to depend on observed rainfall via a carrying capacity term. This is described in subsection”Seasonal parameter variation”. The updated results match intuition, with the additional rainfall term in our model unable to improve model fit, and hence the model with both climate and control still performs better than the climate-only model.

- Why assume sinusoidal variation in temperature (Equation 0.13), given the actual data is available (albeit I accept it looks reasonably sinusoidal). Then the functional form used in Mordecai et al., 2017could be used directly, with priors on those parameters. This would seem to be more satisfying.

1.2 We initially used a sinusoidal function to approximate seasonal variation because this captured much of the observed change in temperature. However, we realise that it would be more rigorous to implement temperature-dependency directly – in the updated manuscript, the vector-specific parameters vary over time based on observed temperature. This approach is described in subsection “Seasonal parameter variation”, and led to the same overall conclusions about which model performed best.

- It would be better to represent the clean-up campaign (which would have reduced larval habitat, not adult mosquito density) as a (perhaps linearly increasing over time) reduction in the mosquito source term (multiplier on recruitment rate δ in Equation 0.10, or of carrying capacity) than a direct modifier of transmissibility. Doing so will offset the impact on transmission by a generation time – i.e. ~2 weeks. It may then be possible to fit a parametrically simpler model to the exact timing of the clean-up campaign.

1.3 There were multiple interventions during the clean-up campaign, including

promotion of awareness about prevention and protection, as well as larval habitat removal. Unfortunately, it is not possible to disentangle these effects with the data available. We therefore included a single term that acts on the vector-to-human transmission rate and could therefore capture both/either of these interventions. We clarify this in subsection “control measures”:

“There were multiple concurrent interventions during the clean-up campaign, including promotion of awareness about protection from bites as well as larval habitat removal. Given the structure of the data available, it would not be possible to independently estimate the extent to which the campaign directly reduced vector-tohuman transmission (i.e. κ(t) acting on α(t)) rather than vector density (i.e. κ(t) acting on m(t)). However, if there was a substantial effect on larval habitat capacity but not on biting rate, we may expect to infer a larger value of a_τ_, to reflect the delay in impact as a result of the time required for vector development.”

We also acknowledge the limitation in the Discussion section:

“We used a flexible time-dependent transmission rate to capture a potential reduction in transmission as a result of control measures in March 2014. The clean-up campaign included multiple concurrent interventions, which occurred alongside ongoing media coverage of the outbreak; it was therefore not possible to untangle how specific actions – such as vector habitat removal or changes in community behaviour that reduced chances of being bitten – contributed to the outbreak decline”

- Little mention is made of the discordance of the MIA assay with the ELISA results – how was seroconversion measured with that assay? If with just a ratio, then perhaps a mixture model would fit better? It would also be good to see the raw results for that assay – the authors spend a lot of time modelling the ELISA results, but then use the MIA data in their default (best fit) model with no comment on the relative reliability of one vs the other. Figure 3B should also be updated to show the MIA results as well as the ELISA ones. More generally, why were two serological assays used? The motivation for this is never stated.

1.4 We have updated the Materials and methods section to clarify the reasons for using different assays:

“Because the indirect IgG ELISA does not distinguish between DENV serotypes, samples were also tested against each of the four specific DENV serotypes using a recombinant antigen-based microsphere immunoassay (MIA), as previously used to examine seroprevalence against different flaviviruses in French Polynesia [20, 30]. Specifically, we wanted to measure the change in seropositivity to DENV-3 during the study period. As an additional validation, a subset of fifty samples from Central Division – including a mixture of those seronegative and seropositive by ELISA and MIA – were tested for the presence of neutralising antibodies against each of the four DENV serotypes using a neutralisation assay as previously described [31]. A neutralisation titre of ≥20 was defined as seropositive (Figure 6—figure supplement 2A).”

We also provide more details about how the different assays compare with one another:

“For both MIA and neutralisation assay results, the largest change in seropositivity was for DENV-3 (Supplementary File 1B). When seropositivity to any DENV (i.e. seropositive to at least one serotype) was compared, a similar change was observed across ELISA, MIA and neutralisation assay results between 2013 and 2015.”

Finally, we show the raw distribution of MIA values, and explain that the distribution of these values means that is not possible to fit a mixture model to such data:

“It was not possible to perform the same analysis using the MIA data because unlike the ELISA and neutralisation assay data, the raw MIA values did not follow a bimodal distribution that indicated likely naive and previously exposed individuals (Figure 6—figure supplement 2B)”

- Table 2 should have the MIA results for 'Any dengue' shown to be able to compare properly with ELISA, esp given the MIA model is the one presented in Figure 5.

1.5 We have now added these results to the table.

- I found it difficult to reconcile the data plotted in Figure 1A with the data points plotted on Figure 5A. Which region was being fitted to – I presumed the whole country, but if that's the case, I don't know why the case numbers peak at about 600 in Figure 5A but at over 2000 in the black curve of Figure 1A. Are weekly case number being plotted in some places, and monthly in others? If so, I suggest using weekly numbers throughout (and updating figures to state 'weekly cases'). It would also be good if Figure 1A was a bit bigger.

1.6 The model in Figure 5 was fitted to Central Division only, with lab tested cases shown. The data shown in Figure 1A was all reported cases (i.e. DLI and lab), which is why the totals were larger. We realise these could have been labelled more clearly: the updated Figure 1A now shows lab tested cases, with a larger figure, to match the data shown in Figure 6. We have also relabelled Figure 5 to clarify what region is fitted.

- It was unfortunate that disease surveillance changed almost coincident with the clean-up campaign. But I was uncomfortable with the authors dropping 2-4 weeks of surveillance data from the fit. The authors refer to a sensitivity analysis for the number of weeks of data dropped, but I couldn't find the results. At the very least, cases of the two types for the missing 2 weeks should be plotted in Figure 5A, shown in a different colors.

1.7 We agree that the change in surveillance during the outbreak poses a challenge for inference. In Supplementary file 3 of the original submitted manuscript we showed a sensitivity analysis for different weeks of data omitted, finding that the number of weeks omitted (between 2–4) did not change our overall conclusion. However, we appreciated the reviewer’s suggestion below of including these additional weeks of data in the fitting, and now do this – details are given in response 1.8 below. We also now show both time series in Figure 5 and the corresponding supplementary figures.

- I think the authors could avoid dropping those critical weeks of data, Assuming the transition occurred at different times for different health facilities, on week t let p_t_ denotes the proportion of surveillance using lab confirmation, and (1-p_t_) be the proportion using DLI. Then the expected total number of reported cases (lab plus DLI) is [p_t_ r_1_ + (1-p) r_2_]c_t_. If O1t and O2_t_ are the reported cases in week t from the two surveillance systems, then O1_t_/O2_t_ gives an estimate of p_t_ /(1-p_t_). This chain of reasoning can be represented in the likelihood in a number of ways – by using separate likelihoods for the lab and DLI case, and estimating p_t_ for each of the 2 currently omitted weeks explicitly, or (more crudely) by pre-calculating p_t_ from O1_t_ /O2_t_ for the transition period, and using a single likelihood for total cases (albeit still with r_1_ and r_2_ – though a single k would need to be used in that case). Or there may be some cleverer way to do it! As an aside, I wasn't quite clear why different k values were fitted to the different case types – were the estimates different?

1.8 Thank you for this suggestion. We have now updated the model to jointly fit to both datasets over the course of the outbreak. Rather than try and calculate the ratio for all weeks, which would not be identifiable, we used the reviewer’s suggestion of pre-calculating p_t_. This is detailed in subsection “Model Fitting”.

The original k values were fitted independently to account for potential differences in reporting variability by type, but in the updated framework, we use the same k so that the cases can be fit jointly, as described above.

- Wouldn't a beta-binomial model (parameterised in terms of a mean p and a overdispersion parameter) be a better representation of the observation process than a negative binomial?

1.9 We used a negative binomial rather than a beta-binomial as it interprets reporting as a rate rather than as a proportion, and hence allows for the weekly variability to potentially generate over-reporting. This is a particularly useful property in the later stages of the outbreak, where case numbers are low. We now mention this in the updated Materials and methods section.

“We used a negative binomial distribution to allow for both under- or over-reporting, the latter being potentially relevant in the final stages of the outbreak when case numbers were low.”

- Any thoughts on why the first DLI point shown in Figure 5A is so high? Though looking at Figure 1A, perhaps this is an artefact of omitting some data points? Otherwise, could it indicate a transiently negative impact of the intervention? How does the model fit change if that point is excluded (see my early ref to the 2-4 week sensitivity analysis mentioned but not presented)?

1.10 This was indeed an artefact of omitting prior data points, but this is no longer an issue in the figure as the updated plot shows both time series.

*- Supplementary File 1 – the EIP is a bit long in my opinion – the blood feeding experiments of Simmons etc. might suggest 10 days is a more reasonable value. Some sensitivity analysis to this would be useful, given changing the generation time will change* R_0_*and thus the relationship between infection attack rates and case incidence (perhaps!)*

1.11 In the updated model, we include temperature dependent EIP based on experimental data including that of Simmons et al., as detailed in response 1.2 above. The expected EIP in our new prior distribution, based on these data, is 10 days when the temperature is 25°C. However, we include some uncertainty in this prior (Figure 5—figure supplement 4), and so if the mean EIP is slightly longer or shorter in reality, this uncertainty will be reflected in the model results.

- Regarding Figure S7 and the need for time-varying transmission neglects individual-level heterogeneity in exposure. I get the basic point about the limited serological attack rate, but heterogeneity could in theory explain this. Such heterogeneity is quite extreme for malaria, and likely to be comparable for dengue, given the more limited vector dispersion range. I accept that time-varying transmission is a more likely explanation, but the authors may still want to comment. In addition, while the authors didn't find significant predictors of seroconversion (unsurprising, given the small sample size), the RRs largely agree with intuition.

1.12 We agree that this point would be worth acknowledging, and now address in the Discussion section:

“Some of the discrepancy between the high attack rate predicted by a randomly mixing model and lower observed seroconversion could in theory be explained by heterogeneity in transmission [18], but the high level of seroprevalence in older age groups suggests that only a small proportion of individuals have consistently avoided infection (Figure 3).”

We also now comment on the risk factors in the Discussion section:

“We did not identify environmental factors that were significantly associated with infection, likely as a result of the relatively small sample size in the serological survey, but the estimated odds ratios were broadly consistent with factors that would be expected to increase or decrease infection risk (Table 3).”

- The model comparison exercise (Supplementary file 7) should be described in the Materials and methods section.

1.13 We have added details of the model comparison to the Materials and methods section.

- Where are all the parameter estimates for the models shown in Table S3? These need to be given. Or at least for the SEIR+climate vs SEIR+climate+control models (ELISA and MIA variants)?

1.14 We have updated the Supplementary Information to include Tables showing parameter estimates for the three models and two serological data types. Full model outputs can also be reproduced using the accompanying code (see response 1.15).

- All data needs to be released with the paper to allow reproducibility of results (i.e. raw data from serosurvey, surveillance data at the resolution used, climate data)

1.15 The data and code required to reproduce the analysis are already available on GitHub (http://github.com/adamkucharski/fiji-denv3-2014), as was noted in the Materials and methods section and on the transparent reporting form. However, we realise we could have been clearer about data availability in the submission information and have made this update.

Reviewer #2:This manuscript combines a mathematical modeling analysis with two empirical data sources pertaining to the 2013-2015 epidemic of dengue in Fiji. One of the most unique aspects of this work is that the authors had the good fortune of having a number of samples collected for other purposes just prior to the epidemic that could be assayed for prior DENV exposure and used to form a longitudinal cohort that was followed up on after the epidemic in 2015. In many cases, epidemic analyses are limited to passive surveillance data, which is also examined here. Thus, this work represents one of a very limited number of opportunities to perform separate and combined analyses with these two different data types, providing insight into the relative strengths and limitations of the two and providing information about the extent of discrepancy between inferences made on one, the other, or both. Moreover, a number of different forms of analysis were conducted, including a mathematical transmission model that was fitted to both data types.Overall, I view this manuscript as having many strengths and believe that it reflects a nice combination of unique data and thoughtful analysis. My primary criticisms of the paper have to do with where emphasis is placed in terms of results, writing, and take-home messages. My impression from reading this manuscript is that some of the primary results in the authors' view pertain to what factors drove the epidemic and their conclusion that vector control made a perceptible contribution to ending the epidemic. While I concede that these claims are plausible and perhaps even likely, I had reservations about the extent of inference drawn based on the analyses that were performed. The transmission model used here is a standard choice, but the reality is that models of this form were devised for theoretical purposes rather than inferential ones. That is not to say that models of this form cannot be used for inference – indeed, I am engaged in work of that nature myself – but there is a great deal of uncertainty in many structural aspects of these models that must be acknowledged and examined before reliable inferences can be drawn. My concern is that the authors have not done enough in that respect.

2.1 We agree that while mathematical models can provide suggestions about potential factors driving the epidemic, such models do have limitations. In our analysis, we tried to be thorough in evaluating potential alternative explanations for observed dynamics. For example, we used DLI data as well as lab tested data, to avoid attributing a decline to control rather than a change in reporting, and jointly fitted to both serology and surveillance data to better examine the role of herd immunity. However, we realise that we could have done more to acknowledge the uncertainty involved and include more plausible assumptions. We have now made such changes, as detailed in responses 2.2–2.8 below.

To elaborate, in the Abstract, it is stated that "Mathematical modelling showed that temperature-driven variation in transmission and herd immunity could not fully explain observed dynamics." The authors then go on to state that there was an additional reduction in transmission explained by a vector clean-up campaign. While they may be right, a problem with these statements here and elsewhere in the manuscript is that the one relatively simple model chosen was used to make a rather conclusive statement about what did or did not drive the observed dynamics (e.g., "mathematical modeling" rather than "a mathematical model" showed).

2.2 We have updated the manuscript to make it clearer that we were fitting with one specific type of model, and that the results presented come from this model. For example, in the Abstract we now state:

“A mathematical model jointly fitted to surveillance and serological data suggested that climate-driven variation in transmission and herd immunity could not fully explain observed dynamics. However, the model showed evidence of an additional reduction in transmission coinciding with a vector clean-up campaign, which may have contributed to the decline in cases in the later stages of the outbreak.”

And in the Discussion section:

“With the addition of serological data in the model fitting, however, our model was able to quantify the relative contribution of herd immunity, climate and control measures to the outbreak dynamics. In particular, this model suggested that seasonal variation and herd immunity alone could fully explain the fall in transmission.”

I can think of numerous ways in which this model could be elaborated on or alternatives proposed (and not necessarily more mechanistic detail but potentially more flexibility from more statistical descriptions in certain places) that would likely better fit the data and could lead to different conclusions. For example, the model did not allow for a dynamic vector population, which could be a major factor in driving seasonal transmission in its own right but also leads to changes in the demographic composition of the vector population that are extremely important epidemiologically and that interact in important ways with control measures. And while there was some allowance for seasonality, it entered the model in a somewhat odd way (effectively influencing biting rate and vector competence but not EIP, mortality, etc.).

2.3 As the implementation of seasonality and vector dynamics was also raised by reviewer 1, we have expanded the model to include temperature and rainfall dependent vector parameters. These additions are described in subsection “Seasonal parameter variation”.

Given how quickly a list of potentially major concerns about the model's structure can be generated (not to mention the very particular assumption of sinusoidal seasonality, which there is no good reason in principle to expect as opposed to other seasonal forms), it may be overreaching to make strong claims about ruling out or supporting certain factors in driving the epidemic. The results related directly to the model are suggestive, but that's about it.

2.4 We have made substantial amendments to model structure and assumptions, including removing the assumptions of sinusoidal seasonality (see response 1.2 above) and adding rainfall and temperature dependent parameters (response 2.3). We have also edited wording to make it clearer that results are dependent on our model (response 2.2). Although there are several limitations our approach, as acknowledged in the Discussion section, we believe the model results still provide useful insights into the outbreak dynamics and illustrate how combining surveillance and serological data can suggest results that would not be obtained with surveillance data alone.

Inferences based on the serological data are also not completely straightforward. First, there is the difference between what the IgG and MIA assays tell you about: infection with one's first serotype and infection with a particular serotype. Second, the model does not account for multiple serotypes and is thus presumably intended as a DENV-3 model only. The surveillance data are reflective of all DENV serotypes, however, so some discrepancy between models fitted to these data sources is to be expected based on that alone. Whether in fitting the model or performing the age analysis, a lot of care must be taken when dealing with data that speak to non-specific seroconversion. The reason is that there could have been people experiencing their second infection but who did not seroconvert because they were already seroconverted by the time of the period of observation. The MIA data address this issue to a large extent, but that seems to be done in an either/or way using IgG or MIA but not both sources of information.

2.5 We agree that it is challenging to interpret ELISA results given potential prior exposure. In our analysis, we therefore used two different approaches to account for this issue.

First, we examined the change in raw ELISA values, rather than just relying on seropositivity. This made it possible to identify individuals who were initially seropositive, but subsequently had evidence of infection. We clarify this in subsection “‘Serological modelling”’:

“Based on seropositivity in 2015 alone, it would not be possible to identify infections during the 2013/14 outbreak among individuals who were initially ELISA seropositive in 2013. We therefore examined the changes in paired individual-level ELISA values between 2013 and 2015.”

We further address the issue of non-specific seroconversion in the Discussion section:

“In addition, if age-specific infection rates are indeed higher in younger groups, it means that estimating population attack rates based on the proportion of seronegative individuals infected may over-estimate the true extent of infection. Focusing on the seronegative subset of the population leads to children being oversampled, which in our data inflates attack rate estimates by around 10% compared to estimates based on change in ELISA value”

Second, we compared model results generated under the assumption that non-specific seropositivity conferred immunity (i.e. using ELISA data) with the assumption that only DENV-3 seropositivity reflected immunity (i.e. using MIA data). Our overall conclusions about the role of climate and control measures were the same (Table 5), indicating that our analysis was not strongly dependent on the assumption used.

These assumptions are now clarified in the Materials and methods section:

“We also fitted the model to either MIA or ELISA seroprevalence data, to reflect two different assumptions about the relationship between seroprevalence immunity. The model using MIA data assumed that only individuals who were seropositive to DENV-3 were immune; the model using ELISA data assumed that seropositivity to any DENV serotype conferred immunity.”

Additionally, I do not have much familiarity with MIA and would have appreciated more exposition about its validity. For example, how do its abilities to infer serotype-specific exposure compare to PRNT? Could some PRNTs be done on samples from this study to demonstrate that? Are results interpretable for individuals with multiple DENV exposurs?

2.6 We now include a comparison with results from neutralisation assay performed on a subset of 50 samples. The results of this validation are described in the updated Materials and methods section:

“For both MIA and neutralisation assay results, the largest change in seropositivity was for DENV-3 (Supplementary file 1B). When seropositivity to any DENV (i.e. seropositive to at least one serotype) was compared, a similar change was observed across ELISA, MIA and neutralisation assay results between 2013 and 2015.”